# Derepression of the epithelial transcription factor GRHL2 promotes direct hepatocyte-to-cholangiocyte transdifferentiation

Ludivine Vasseur[1], Céline Gheeraert[1☺], Julie Dubois-Chevalier[1☺], Ninon Very[1], Loïc Guille[1], Mohamed Bou Saleh[2], Clémence Boulet[1], Cyril Sobolewski[2], Pascal Loyer[3], Alexandre Berthier[1], Noémie Legrand[2], Anne Corlu[3], Viviane Gnemmi[4], Guillaume Lasailly[2], Emmanuelle Leteurtre[4], Dmitry Galinousky[1], Antonino Bongiovanni[5], Solenne Taront[2], Nicolaj I. Toft[6], Lars Grøntved[6], David Tulasne[7], Alessandro Furlan[7,8], Line Carolle Ntandja-Wandji[2], Bart Staels[1], Philippe Lefebvre[1], Laurent Dubuquoy[2], Jérôme Eeckhoute[1]*

1 Univ. Lille, Inserm, CHU Lille, Institut Pasteur de Lille, U1011-EGID, F-59000, Lille, France, 2 Univ. Lille, Inserm, CHU Lille, U1286 – INFINITE – Institute for Translational Research in Inflammation, F-59000, Lille, France, 3 Inserm, INRAE, Univ Rennes, Institut NUMECAN (Nutrition Métabolismes et Cancer) UMR-A 1341, UMR-S 1317, F-35000, Rennes, France, 4 Univ. Lille, CNRS, Inserm, CHU Lille, UMR9020-U1277 – CANTHER – Cancer Heterogeneity Plasticity and Resistance to Therapies, F-59000, Lille, France, 5 Univ. Lille, CNRS, Inserm, CHU Lille, Institut Pasteur de Lille, US 41 – UAR 2014 – PLBS, Lille, France, 6 Department of Biochemistry and Molecular Biology, University of Southern Denmark, Odense M, Denmark, 7 Univ. Lille, CNRS, Inserm, CHU Lille, Institut Pasteur de Lille, UMR9020 – U1277 – Canther – Cancer Heterogeneity, Plasticity and Resistance to Therapies, Lille, France, 8 Tumorigenesis and Resistance to Treatment Unit, Centre Oscar Lambret, F-59000, Lille, France

☺ These authors contributed equally to this work.
* jerome.eeckhoute@inserm.fr

## Abstract

The liver's regenerative capacity is underscored by the plasticity potential of adult hepatocytes. In this context, hepatocyte-to-cholangiocyte transdifferentiation (HCT) has been ascribed with pro-regenerative functions in animal models and is a feature of end-stage human chronic liver diseases. While dampened activities of hepatocyte identity transcription factors (TFs) underlay HCT, how the cholangiocyte transcriptional program is implemented is poorly defined. Here, we identify that HCT does not involve transitioning through a hepatoblast-like transcriptional program. Furthermore, we show that HCT primarily involves induction of the archetypal transcriptional program of monopolarized epithelial cells initially repressed in hepatocytes. Indeed, HCT requires relieving H3K27me3-mediated and polycomb-dependent epigenetic silencing of epithelial TF encoding genes including *Grainyhead Like Transcription Factor 2* (*GRHL2*). Ectopic expression of GRHL2 in hepatocytes, including in vivo in the adult mouse liver, induces epithelial genes reminiscent of those activated during HCT. Finally, GRHL2 is detected in human hepatocytes undergoing HCT as evidenced using samples from end-stage chronic liver diseases. Hence, HCT is a process chiefly characterized by induction of a conventional epithelial transcriptional program

**Data availability statement:** All individual numerical values that underlie originally generated data in this study data are available in the S1 Data file. All ChIP-seq and RNA-seq data generated in this study have been deposited into the Gene Expression Omnibus SuperSeries under accession number GSE281717. Codes and data related to integrated mouse liver single-cell RNA-seq data have been deposited into zenodo (https://zenodo.org/records/13897656 and https://zenodo.org/records/13897729, respectively).

**Funding:** This work was supported by the Association Française pour l'Etude du Foie (AFEF to NV), Fondation de l'Université de Lille (to MBS), Fondation pour la Recherche Médicale (FRM grants EQU202203014645 to PL, EQU202003010299 to BS and EQU20200310299 to LD) and the Agence Nationale de la Recherche (ANR) grants "HSCreg" (ANR 21-CE14-0032-01 to JE) and "MEdicAL" (ANR 21 CE17-0016-02 to LD) and "European Genomic Institute for Diabetes" E.G.I.D. (ANR 10 LABX-0046), a French State fund managed by ANR under the frame program Investissements d'Avenir I-SITE ULNE/ANR-16-IDEX-0004 ULNE. This work also received support from Métropole Européenne de Lille (MEL) and Université de Lille (Investissements d'Avenir I-SITE ULNE/ANR-16-IDEX-0004 ULNE) (V-Chaire Industrielle-23-003 to JE). LV received support from EGID (ANR 10 LABX-0046), PreciDIAB (ANR-18-IBHU-0001) and Conseil Régional des Hauts-de-France (convention 22005973) as well as from Fondation Recherche Médicale (FDT202504020162). The funders had no role in study design, data collection and analysis, decision to publish, or preparation of the manuscript.

**Competing interests:** The authors have declared that no competing interests exist.

**Abbreviations:** AAV3, adeno-associated viruses serotype 3; AAV8, adeno-associated viruses serotype 8; ALB, albumin; ALD, alcohol-related liver diseases; ALGS, Alagille syndrome; APAP, acetaminophen; BMEL, bipotential mouse embryonic liver; ChIP, chromatin immunoprecipitation; DAVID, database for annotation, visualization, and integrate

originally lacking in hepatocytes promoted by derepression of the master epithelial TF GRHL2.

## Introduction

Hepatocytes, the predominant liver cell type, play a crucial role in the organ's homeostatic and secretory functions. Indeed, hepatocytes are key regulators of energy and iron homeostasis, and bear critical functions related to detoxification, coagulation, and the acute phase response. These unique hepatocyte functionalities are acquired during cellular differentiation, which is itself driven by establishment of a hepatocyte-specific transcriptional program. This phenomenon is orchestrated by transcriptional regulators, primarily hepatocyte identity transcription factors (Hep-ID TFs), which are hepatocyte-specific/enriched TFs [1–3].

Hepatocytes originate from hepatoblasts, which can alternatively differentiate into cholangiocytes. Cholangiocytes line the lumen of bile ducts and control bile volume and composition [4]. Together, hepatocytes and cholangiocytes constitute the two types of epithelial cells found in the liver [5]. However, unlike cholangiocytes, which form a conventional monolayer of cuboidal or columnar epithelial cells with a single apico-basal axis (hereafter defined as monopolarized epithelial cells), hepatocytes are characterized by multiple apical (and basal) surfaces, a unique feature among epithelia [6–8]. The liver possesses unique regenerative capacities where epithelial cell plasticity is instrumental while the existence of adult liver stem cells is highly disputed [9–11]. Indeed, the plasticity of adult liver epithelial cells allows them to reciprocally transdifferentiate [9,10]. In the mouse, this has been identified as a pro-regenerative mechanism operating when repair through proliferation of the injured cell type is not sufficient and/or impaired [9].

For example, in a mouse model mimicking the phenotype of Alagille syndrome (ALGS) with a deficient intrahepatic biliary system at birth, hepatocytes transdifferentiate to fully mature cholangiocytes to form functional bile ducts [12]. Lineage tracing has provided compelling evidence for transdifferentiation of liver epithelial cells in such conditions [12]. In humans, end stages of chronic liver diseases, such as alcohol-related liver diseases (ALD) [13] and metabolic dysfunction-associated steatotic liver disease (MASLD) [11], are characterized by the presence of cells co-expressing hepatocyte and cholangiocyte marker genes. In line, transcriptomic analyses of hepatocytes from humans with end-stage ALD and MASLD revealed instances of hepatocyte-to-cholangiocyte-like cell transdifferentiation [11,14]. HCT is also observed in livers from patients with primary sclerosing cholangitis (PSC) [15]. In this context, hepatocyte-derived cells contribute to the ductular reaction, a histologically defined phenomenon of hyperplasia of bile duct-like structures [16].

Despite its pathophysiological importance, HCT remains a poorly understood phenomenon. In particular, how HCT is orchestrated at the molecular level to induce a cholangiocyte-like transcriptional program remains ill-defined.

discovery; DDC, 3,5-diethoxycarbonyl-1,4-dihydrocollidine; DHS, DNase I Hypersensitivity; DMSO, dimethyl sulfoxide; DTT, dithiothreitol; GRHL2, Grainyhead Like Transcription Factor 2; GSEA, gene set enrichment analysis; HBE, human bronchial epithelial; HCT, hepatocyte-to-cholangiocyte transdifferentiation; Hep-ID TFs, hepatocyte identity transcription factors; MASLD, metabolic dysfunction-associated steatotic liver disease; MPH, mouse primary hepatocytes; NPC, non-parenchymal cells; nTPM, normalized gene expression in transcripts per million; PBC, primary biliary cholangitis; PBS, phosphate-buffered saline; PCA, principal component analysis; PFA, paraformaldehyde; PHx, partial hepatectomy; PRC2, polycomb repressive complex 2; PSC, primary sclerosing cholangitis; qPCR, Quantitative PCR; RT, room temperature; scRNA-seq, single-cell RNA-seq; SDS, sodium docetyl sulfate; TBS, Tris-Buffered Saline; TFs, transcription factors; TSS, transcriptional start sites; UMAP, uniform manifold approximation and projection.

## Results

### HCT occurs directly without transitioning through a hepatoblast-like transcriptional state

In several previous studies, HCT was proposed to involve dedifferentiation of hepatocytes into so-called bipotent liver progenitor-like cells [17–19]. To further compare gene expression changes involved in HCT with those defining developmental cholangiocyte differentiation from progenitors, we integrated single-cell RNA-seq (scRNA-seq) data obtained from embryonic, postnatal, and adult mouse livers [20,21]. In order to broadly capture the hepatocyte heterogeneity landscape, data from adult mouse livers comprised a wide range of injury models: acute exposure to acetaminophen (APAP) [22,23] or lipopolysaccharide [24], chronic exposure to 3,5-diethoxycarbonyl-1,4-dihydrocollidine (DDC) inducing cholestasis [25,26], western diet-induced metabolic dysfunction-associated steatohepatitis [27] and partial hepatectomy (PHx) [23,28,29] (S1 Table). Cells from the liver epithelial lineage were recovered from each dataset and batch correction was performed before visualization using uniform manifold approximation and projection (UMAP) (see Materials and methods for details). This approach resulted in the segregation of hepatoblasts from mature cholangiocytes or hepatocytes (Figs 1A and S1A). For instance, clustering analysis identified a cluster of cholangiocytes (denoted cluster 1; S1B Fig). Interestingly, this cluster was, however, connected to two other clusters comprising, on the one hand, hepatoblasts/postnatal epithelial cells (cluster 2) and, on the other hand, mostly adult hepatocytes (cluster 3) (Figs 1B and S1B). Interrogating the origin of hepatocytes from cluster 3 revealed that they almost exclusively stemmed from DDC-treated mice (95%; Fig 1C) including yellow fluorescent protein-labeled hepatocytes used for lineage tracing in this study [25] (S1C Fig). This is consistent with cluster 3 capturing HCT, which is known to be prominently induced in the chronic DDC-induced cholestasis model, most probably due to exhaustion of the cholangiocyte regenerative potential [9,30]. These data also indicated that HCT was unrelated to hepatocyte reprogramming occurring in other conditions such as APAP intoxication or PHx. This implies that induction of SOX9 in a subset of hepatocytes, commonly observed across these injury models (S1D Fig) and interpreted in the literature as hepatocytes acquiring biliary features [31], is in fact associated with different context-specific hepatocyte reprogramming events not necessarily involving transition towards a cholangiocyte-like cellular state. Using cells from clusters 1, 2, and 3, we performed RNA velocity analyses, which indicated that the developmental differentiation of cholangiocytes either from hepatoblasts or through HCT are independent (Figs 1D and S1E). In line, refined analysis of cellular trajectory and fate transition probability using CellRank2 [32] indicated that a subset of hepatocytes from DDC-treated mice (called HCT) was characterized by an extremely high probability to become cholangiocytes (Fig 1E and 1F). Further arguing against HCT involving hepatocytes to transiently adopt a progenitor-like transcriptional program, no hepatocytes from DDC-treated mice displayed significant fate probability towards hepatoblasts (Figs 1F and S1E). Finally, the fraction of variable genes showing peak expression at intermediate stages during HCT was limited (9%; S1F Fig). Moreover, these genes

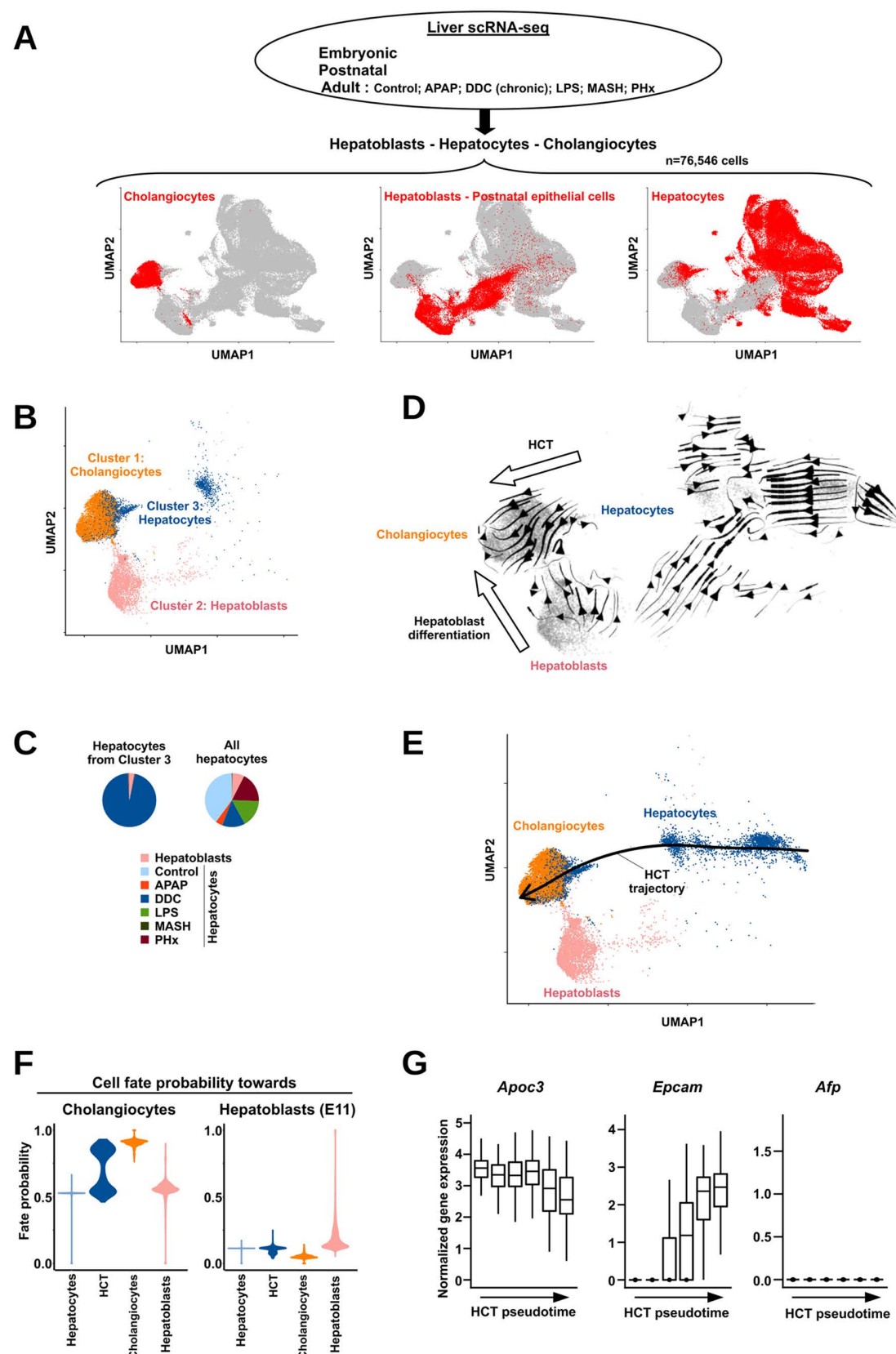

**Fig 1. Characterization of HCT using a scRNA-seq atlas of mouse parenchymal liver cells. (A)** Overview of datasets used to generate our mouse liver scRNA-seq atlas together with the UMAP projection of the obtained 76,546 individual cells and positioning of hepatoblasts (embryonic and postnatal cells), hepatocytes and cholangiocytes. Congruent expression of marker genes is shown in S1A Fig. **(B)** Data from A were used for cell clustering (S1B Fig). Shown here is a zoom of the UMAP projection highlighting the 3 clusters selected to further study HCT vs. developmental cholangiocyte differentiation. The indicated main cell type in clusters 1-3 stems from analyses shown in S1C Fig. **(C)** Pie charts showing the sample of origin of hepatocytes found in cluster 3 compared to all hepatocytes originally analyzed (panel A). **(D)** Results of RNA velocity analyses showing the cell transitions associated respectively with developmental differentiation of cholangiocytes from hepatoblasts and HCT. A magnification of the area containing cholangiocytes is shown in S1E Fig. **(E)** HCT trajectory inferred from pseudotime analyses is shown on the UMAP from panel A. The indicated directionality is in line with the RNA velocity data shown in panel D. **(F)** Violin plots showing the cell fate transition probabilities of individual cells from the indicated groups towards adult cholangiocytes (left) or towards hepatoblasts (embryonic day 11; right) obtained using CellRank [142]. HCT refers to hepatocytes from mice chronically exposed to DDC. **(G)** Boxplots showing the normalized expression of the indicated marker genes in discrete hepatocyte cell subsets along the course of HCT as defined using pseudotime analysis (see Materials and methods for details).

were enriched for biological pathways shared with both hepatocytes and cholangiocytes (S1G Fig) and were not characterized by heightened expression in hepatoblasts (S1H Fig). Indeed, hepatocytes in intermediate stages in the HCT process still express hepatocyte markers [e.g., *Apoc3* (Apolipoprotein C-III) or *Alb* (Albumin)] and start expressing cholangiocyte markers [e.g., *Epcam* (*Epithelial cell adhesion molecule*)]) while lacking expression of typical progenitor genes [e.g., *Afp* (*Alpha fetoprotein*)] (Figs 1G and S1I).

Altogether, these data indicate that HCT occurs through a process, which does not involve transient acquisition of a progenitor-like cell state.

## HCT is mainly characterized by induction of the transcriptional signature of monopolarized epithelial cells in hepatocytes

In order to characterize the transcriptional reprogramming underlying HCT, we compared the transcriptome of lineage-traced hepatocytes from DDC-treated mice with that from healthy hepatocytes and cholangiocytes obtained using bulk RNA-seq of sorted cells (see Materials and methods) [25,33]. When projected onto a space defined by the first two principal components from a principal component analysis (PCA) conducted on the transcriptomes of hepatocytes and cholangiocytes, hepatocytes undergoing HCT localized near cholangiocytes (Figs 2A and S2A). As expected, these data therefore verified that hepatocytes undergoing HCT in the liver of DDC-treated mice have a transcriptome similar to that of cholangiocytes. In line, gene set enrichment analysis (GSEA) indicated that genes characterized by enriched expression in cholangiocytes among the different liver cell-types ($n = 57$ obtained from [34] and denoted as Chol-enriched genes; S2 Table) were significantly biased towards up-regulated genes during HCT (Fig 2B). To unbiasedly assess the nature of the transcriptional reprogramming, enrichment analyses for biological or molecular pathways and their clustering by similarity was next performed using the database for annotation, visualization, and integrate discovery (DAVID) [35]. In line with a transdifferentiation process, genes down-regulated in hepatocytes undergoing HCT, when compared to healthy hepatocytes, were enriched for terms related to normal hepatocyte functions (S2B Fig). When up-regulated genes were considered, enriched terms pointed to archetypal epithelial cell features such as cilium and tight junctions (Figs 2C and S2C). GSEA further indicated that genes characterizing the human epithelial cell lineage identified in [36] ($n = 837$; hereafter denoted as Epith-Signature. See S2 Table) displayed significantly greater expression in hepatocytes undergoing HCT (Fig 2D). Similar results were obtained when using genes characteristic of the monopolar cell polarity (S2D Fig). Moreover, genes from the Epith-Signature were enriched in the subset of HCT up-regulated genes characterized by low basal expression in healthy hepatocytes and strong induction in HCT ($p$-value $< 0.0001$, chi-squared test; Fig 2E). Among those genes were commonly used cholangiocyte markers such as *Krt7* (*Keratin 7*), *Krt19* (*Keratin 19*), and *Epcam* (Fig 2E; over 60% of the Chol-enriched genes are comprised within the Epith-Signature). Consistent with these findings, comparing expression of genes from the Epith-Signature in different human cells from the epithelial lineage obtained from the Human Protein Atlas [37] indicated that, unlike cholangiocytes, hepatocytes were characterized by low expression levels

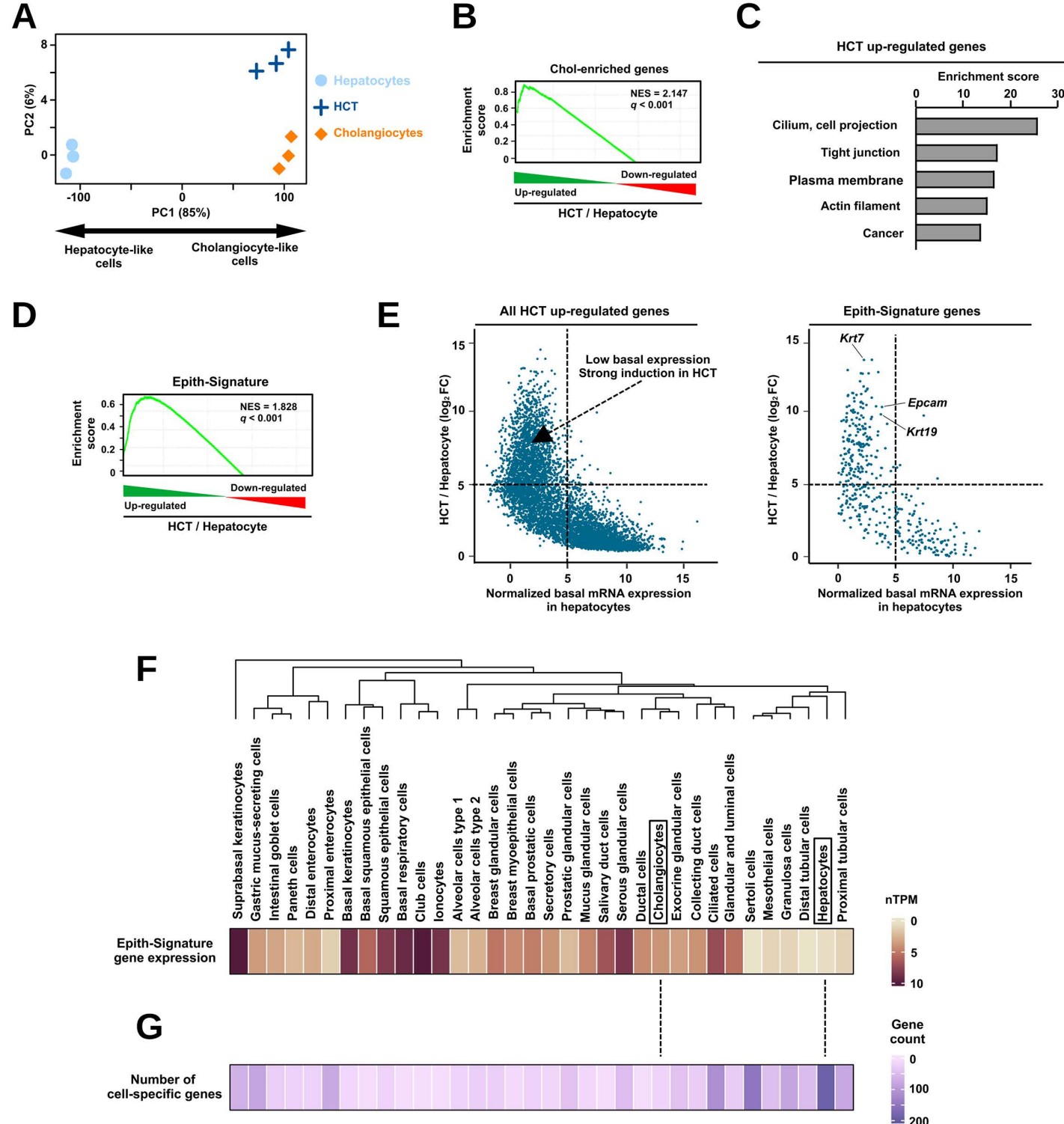

**Fig 2. Activation of a *bona fide* epithelial gene transcriptional program characterizes HCT. (A)** Transcriptomic changes induced in HCT were assessed by projecting datasets obtained from lineage-traced hepatocytes undergoing HCT from DDC-treated mice (denoted HCT) within a 2-dimensional space generated through a PCA performed using the transcriptome of healthy mouse cholangiocytes and hepatocytes (see the Materials and methods section for details). As indicated, the first principal component (PC1) is sufficient to separate hepatocyte-like from cholangiocyte-like cells as indicated. **(B)**

Enrichment plot obtained using GSEA performed with the Chol-enriched genes [34] as the gene set and transcriptomic changes induced by HCT obtained by comparing bulk RNA-seq data from lineage-traced hepatocytes undergoing HCT from DDC-treated mice compared to that of control hepatocytes from healthy livers from [25]. NES stands for normalized enrichment score. **(C)** Functional annotation of genes up-regulated during HCT ($q$-value<0.05; top 3,000 genes were further selected) performed using the DAVID tool [35]. Deregulated genes were obtained by mining data from [25] as described for panel B. Enriched terms were clustered by the DAVID tool and were named according to words repetitively used in the name of individual enriched terms within each cluster (S2C Fig). Top 5 clusters are displayed together with their enrichment scores (i.e., best enrichment score among individual terms within each cluster). **(D)** Enrichment plot obtained using GSEA performed with the Epith-Signature [36] as the gene set and transcriptomic changes induced by HCT defined as in panel B from [25]. NES stands for normalized enrichment score. **(E)** HCT up-regulated genes (left) or genes from the Epith-Signatures (right) were plotted according to their basal expression levels in healthy hepatocytes and their induction upon HCT. Dotted lines were used to highlight a subset of genes displaying low basal expression and strong induction during HCT. Commonly used cholangiocyte markers (i.e., *Krt7*, *Krt19*, *Epcam*) are individually highlighted. **(F)** Expression of genes from the Epith-Signature [36] was used to perform hierarchical clustering of the indicated individual human epithelial cell types (S2E Fig). The dendrogram issued from these analyses is displayed on the left of the heatmap, which shows the median expression of genes from the Epith-Signature in each individual epithelial cell types. Cholangiocytes and hepatocytes are highlighted using frames. nTPM stands for normalized transcripts per million. **(G)** Heatmap showing the number of genes specifically expressed in a given cell type among all epithelial cell types as defined using a stringent tau specificity index cut-off, i.e., tau>0.9. Ranking of epithelial cell types is similar to that of panel F.

(Figs 2F and S2E). Further indication that the transcriptome of hepatocytes differs from that of other epithelial cells, including cholangiocytes, was provided when mining the number of genes with high expression selectivity among individual epithelial cells [i.e., genes with a tau specificity index >0.9 when all individual epithelial cell types are considered [38]. Indeed, hepatocytes were characterized by the greatest number of specifically expressed genes, i.e., with low expression in the other individual epithelial cells, while cholangiocytes had very few (Fig 2G).

Altogether, these data indicate that a main event underlying HCT is induction of a conventional monopolarized epithelial cell gene signature, which is typically absent in healthy hepatocytes.

### Epithelial lineage TFs are repressed by the polycomb repressive complex 2 in hepatocytes

Mining transcriptional changes underlying HCT using a procedure similar to Fig 2A but applied only to TF-encoding genes generated similar results (Figs 3A and S3A). This revealed that changes in TF-encoding gene expression were fully informative with regards to capturing HCT, which prompted us to further characterize deregulated TFs as a mean to define drivers of HCT. Calling differentially expressed genes ($q$-value<0.05) in hepatocytes that underwent HCT compared to healthy ones identified 115 down- and 423 up-regulated TF-encoding genes, respectively (S3 Table). As expected, down-regulated TFs comprised many Hep-ID TFs (S3B Fig) consistent with loss of hepatocyte identity upon HCT. With regards to up-regulated TFs, 82% of them were also significantly induced in cholangiocyte-like cells derived from hepatocytes in the mouse model of ALGS lacking an intrahepatic biliary system [12] (Fig 3B), consistent with a role for these identified up-regulated TFs in HCT.

The up-regulated TFs displayed low basal expression levels in healthy hepatocytes, and were also the most strongly induced upon HCT (Fig 3C). These data suggest that a subset of up-regulated TFs might be originally repressed in hepatocytes. We therefore next characterized the promoters of up-regulated TFs in healthy hepatocytes by monitoring levels of chromatin accessibility [DNase I Hypersensitivity (DHS)] as well as levels of modified histones prototypically linked to gene activation [histone H3 lysine 4 trimethylation (H3K4me3) and lysine 27 acetylation (H3K27ac)], transcription (H3K36me3 and RNA Polymerase II) or repression (H3K27me3). We used DHS-seq and ChIP-seq data obtained from healthy mouse livers [hepatocytes overwhelmingly contribute to chromatin-based signals obtained using whole liver [39–41] or isolated mouse primary hepatocytes (MPH) (S4 Table). Using the Spark software [42], we identified 3 main clusters of promoters with divergent epigenetic patterns (Fig 3D). The largest cluster (C1) comprised promoters already active in normal liver with accessible chromatin and strong H3K4me3 and H3K27ac signals, while C2 and C3 gradually displayed features of inactive but transcriptionally competent or repressed promoters, respectively. In particular, promoters within C2 displayed moderate levels of both H3K4me3 and H3K27me3, i.e., potential bivalent promoters [43], while promoters in C3 were characterized by lack of any feature of gene activation concomitant with strong repressive H3K27me3 signals (Figs 3D and S4A). Mining single-nucleus ATAC sequencing data [44] confirmed the differences in

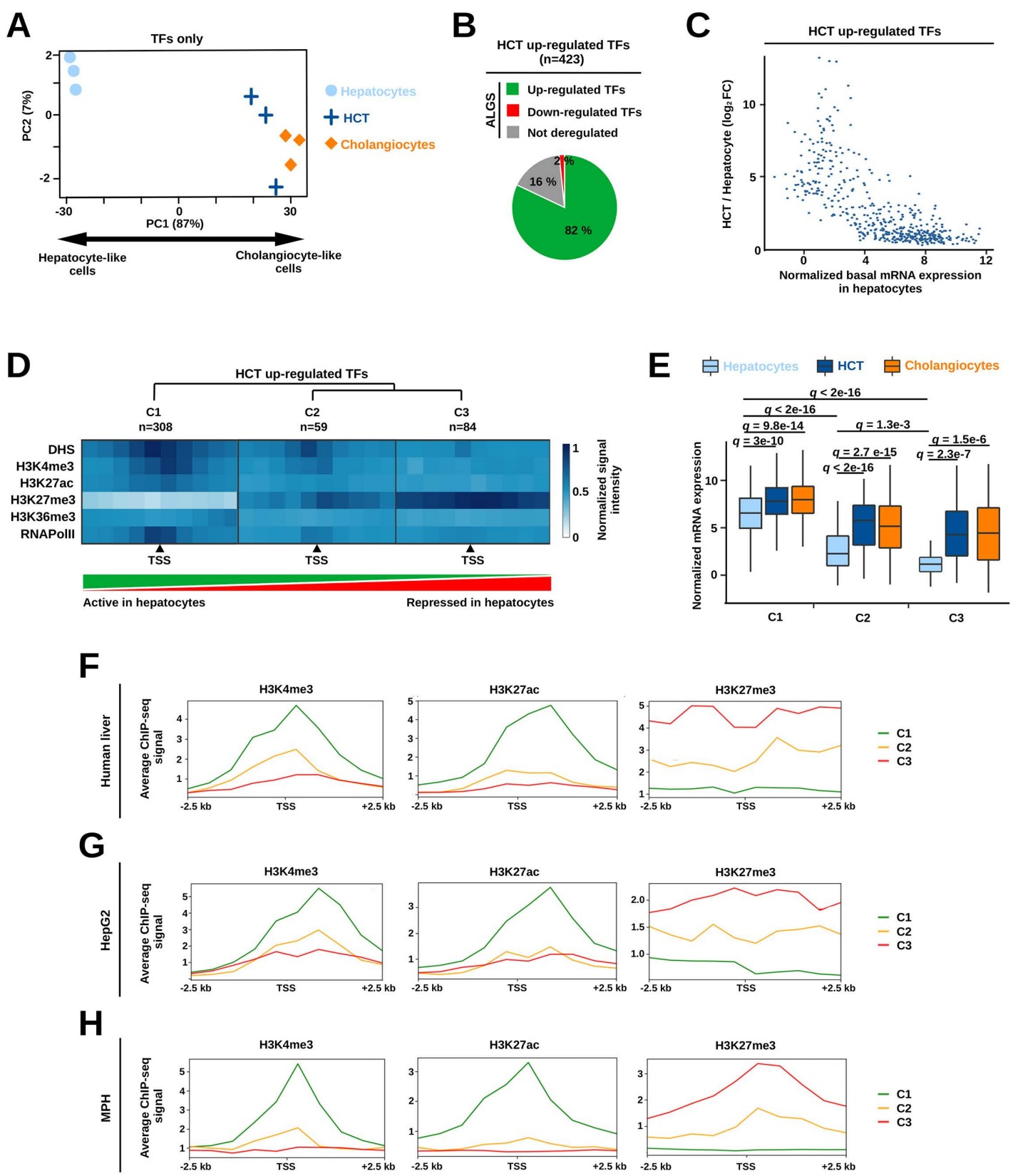

PLOS Biology

**Fig 3. HCT involves induction of TF encoding genes epigenetically repressed in normal hepatocytes. (A)** Analyses performed as in Fig 2A except only TF-encoding genes were considered. **(B)** TF-encoding genes up-regulated during HCT were obtained by comparing bulk RNA-seq data from lineage-traced hepatocytes undergoing HCT from DDC-treated mice compared to that of control hepatocytes from healthy livers from [25] (q-value < 0.05 and $\log_2$ FC > 0). The obtained gene list was monitored for changes in hepatocytes undergoing HCT in another mouse model, i.e., model of ALGS lacking an intrahepatic biliary system at birth [12]. In this second dataset, genes with q-value < 0.15 and $\log_2$ FC < 0 or > 0 were defined as down- or up-regulated, respectively. Genes with q-value > 0.15 were considered as "Not deregulated". **(C)** Induction of TF-encoding genes upon HCT was compared to their basal expression levels in healthy mouse hepatocytes. A scatter plot showing individual TF-encoding genes is shown. **(D)** Average signal for chromatin accessibility (DHS) and indicated modified histones or RNA polymerase II (RNApolII) at TF-encoding gene promoters for the main 3 clusters identified by Spark [42]. The number of TF-encoding genes comprised in each cluster (C1 to C3) is indicated. Dendrogram of hierarchical clustering is also shown. The arrowheads indicate the position of the transcriptional start site (TSS) at the center of the 5 kb windows used for these analyses. **(E)** Box plot showing expression of TF-encoding genes from clusters C1–3 in mouse healthy hepatocytes or cholangiocytes as well as hepatocytes undergoing HCT from [25]. Statistical significance was assessed using two-way ANOVA with Dunnett's multiple comparison post hoc test. **(F–H)** Average ChIP-Seq signals for indicated modified histones at promoters from clusters C1–3 (defined in panel D) in MPH **(E)**, human liver **(F)**, or HepG2 cells **(G)**. Low signal was arbitrarily set to 1. Signals are shown in windows of ± 2.5 kb around the genes TSS.

promoter accessibility between the 3 clusters and did not reveal any difference between periportal and pericentral hepatocytes (S4B Fig). In line with the chromatin status of their promoters, TF-encoding genes from C2, and even more so from C3, displayed low basal expression in mouse hepatocytes but showed greater induction during HCT when compared to those from C1 (Fig 3E). Adding ChIP-seq data for the heterochromatin mark H3K9me3 to Spark did not significantly modify the clustering of up-regulated TF promoters (S5A Fig). In line, genes from C3 were not found within mouse liver chromatin domains primarily characterized by the presence of H3K9me3 [45] (S5B Fig). Consistently, promoters from C3 did not display DNA methylation, which is often found at H3K9me3 repressed genes [46] (S5C Fig). Biophysically identified heterochromatin regions in mouse hepatocytes, defined through sequencing of sonication-resistant heterochromatin [47], indicated that C3 promoters were found both in euchromatin and heterochromatin (S5D Fig). These findings further emphasize the role of H3K27me3 in repressing genes of alternative cell fates in these two chromatin compartments, independently of H3K9me3 [47,48].

We next aimed to define whether observations in mice are consistent with promoter epigenetic patterns found in humans using ChIP-seq data for H3K4me3, H3K27ac and H3K27me3 from human healthy livers [49] and human HepG2 cells [this study and [50]; S4 Table]. We found epigenetic patterns in human livers (Fig 3F) and HepG2 cells (Fig 3G) reminiscent of those observed in mouse hepatocytes (Fig 3D and 3H), which was in line with greater mRNA expression of C1 genes (S6A and S6B Fig). These data both verified the robustness of our clusters of TF-encoding gene promoters and indicated its human relevance.

Deposition of H3K27me3 is under the control of the polycomb repressive complex 2 (PRC2) and more particularly of the enzymes EZH1 and 2 (Enhancer Of Zeste 1 and 2 Polycomb Repressive Complex 2 Subunits) [51,52]. In line, mining ChIP-seq data from [53] (S4 Table) indicated that presence of H3K27me3 at TF-encoding gene promoters in mouse liver is linked to stronger recruitment of EZH2, especially for promoters from C3 (Fig 4A). Moreover, *Ezh1/2* deletion in mouse livers [47] (S4 Table) triggered both a decrease in H3K27me3 (Fig 4B) at the C2 and C3 genes and their transcriptional induction (Fig 4C). This translated into induction of the Epith-Signature and Chol-enriched gene sets (Fig 4D). Finally, treatment of HepG2 cells with pharmacological inhibitors of EZH1/2 (UNC1999 and valemetostat tosylate together referred to as EZHi), which decreased H3K27me3 levels (Fig 4E), also triggered concomitant derepression of TF-encoding genes from C2 and C3 as measured by RNA-seq analysis (Fig 4F).

Altogether, these data indicate that HCT involves the induction of TF-encoding genes normally repressed through polycomb-dependent H3K27me3 in hepatocytes.

## Forced expression of the epithelial lineage TF GRHL2 promotes HCT

We next aimed to demonstrate a functional role for up-regulated TFs in HCT. In addition to being repressed in normal hepatocytes and highly induced upon HCT, TF-encoding genes from C3 displayed the highest tau lineage-specificity

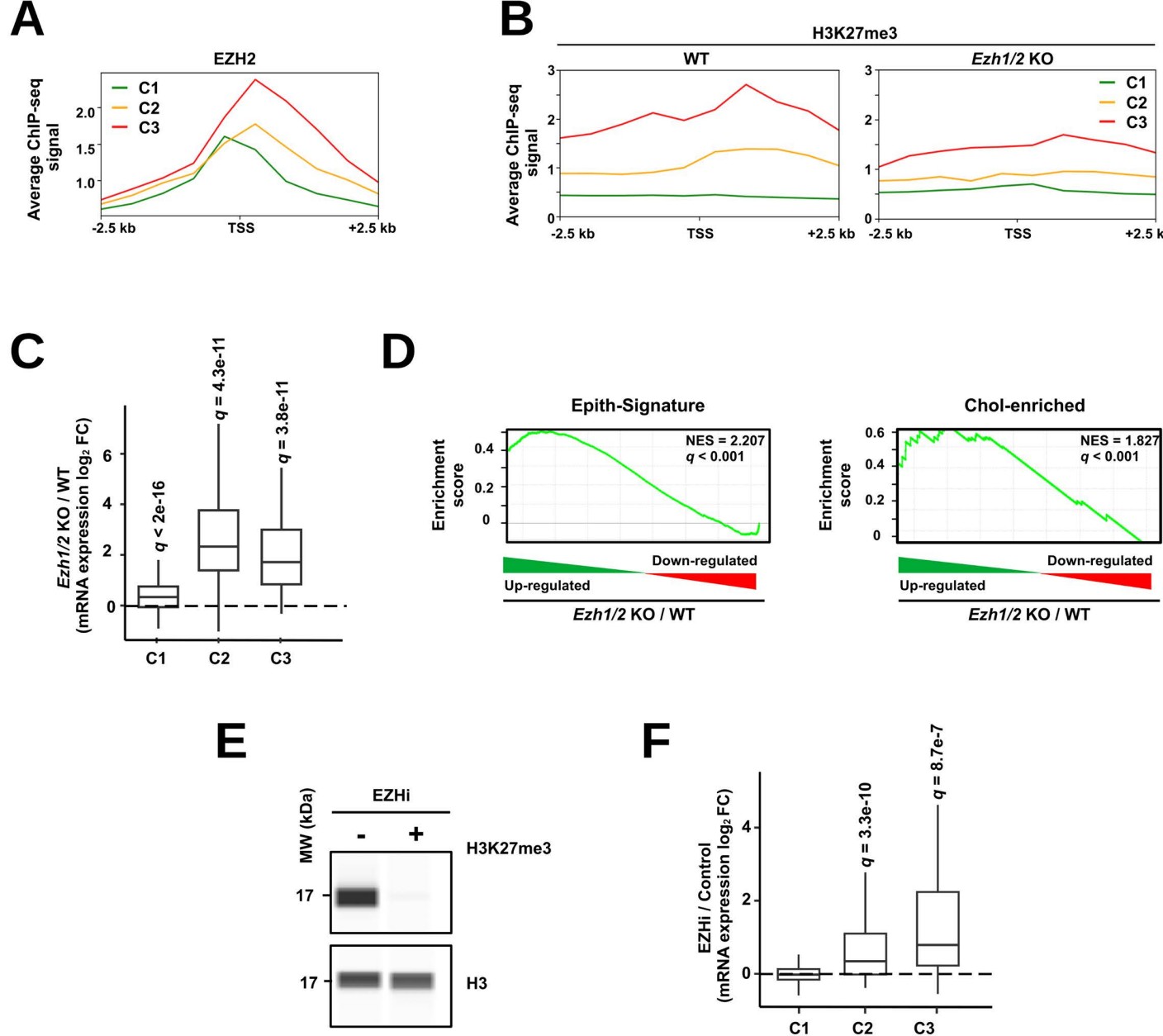

**Fig 4. Polycomb-mediated repression of TF encoding genes up-regulated during HCT. (A)** Average EZH2 ChIP-Seq signal at promoters from clusters C1–3 (defined in Fig 3) in the mouse liver. Low signal was arbitrarily set to 1. Signal is shown in a window of ± 2.5kb around the genes TSS. **(B)** Average H3K27me3 ChIP-Seq signal at promoters from clusters C1–3 (defined in Fig 3) in the liver of wild-type (WT) mice or mice deleted for the *Ezh1* and *Ezh2* genes (*Ezh1/2* KO). Signal is shown in a window of ± 2.5kb around the genes TSS. **(C)** Box plot showing transcriptional modulation of TF-encoding genes expression for clusters C1–3 (defined in Fig 3) obtained from RNA-seq performed on the livers of mice with deletion of both the *Ezh1* and *Ezh2* genes (*Ezh1/2* KO; $n=3$) and wild-type mice (WT; $n=2$) as the control [47]. One-sided one-sample Wilcoxon $t$ test with Benjamini–Hochberg correction was used to determine if the median $\log_2$ FC was significantly above 0, i.e., any statistically significant increase in gene expression between livers of *Ezh1/2* ko and WT mice. **(D)** Enrichment plots obtained using GSEA performed with the Epith-Signature [36] or Chol-enriched genes [34] as the gene sets and transcriptomic changes induced by *Ezh1/2* deletion in the mouse liver (similar to panel C). NES stands for normalized enrichment score. **(E)** Immunoblotting performed using the Wes system to monitor levels of H3K27me3 and H3 in HepG2 cells treated or not with EZH1/2 inhibitors (EZHi). Data are representative of those obtained in 3 independent biological replicates. MW, molecular weight. **(F)** Box plot showing transcriptional modulation of TF-encoding genes expression for clusters C1–3 (defined in Fig 3) obtained from RNA-seq performed on HepG2 cells treated with EZH1/2 inhibitors or 0.2% DMSO as the control ($n=4$ biological independent replicates). One-sided one sample Wilcoxon $t$ test with Benjamini–Hochberg correction was used to determine if the median $\log_2$ FC was significantly above 0, i.e., any statistically significant increase in gene expression between EZHi-treated and control cells. The original data underlying this figure can be found at the Gene Expression Omnibus (GSE281717).

indexes calculated from gene expression in 13 human cell lineages (including the epithelial lineage) obtained from [37] (Fig 5A). We therefore reasoned that C3 comprises key TFs involved in the establishment of the alternative *bona fide* epithelial transcriptional program characterizing HCT. We therefore monitored individual TFs from C3 by simultaneously comparing: *i)* their expression enrichment in epithelial cells (ranking based on highest to lowest expression levels among aforementioned human cell lineages); *ii)* their level of induction upon HCT and *iii)* their presence in the Epith-Signature and Chol-enriched gene set. The TFs Grainyhead Like Transcription Factor 2 (GRHL2) and EHF (ETS Homologous Factor) emerged from these analyses as they were enriched in epithelial cells and cholangiocytes and strongly induced in HCT (Fig 5B). Interestingly, these TFs, together with OVOL1/2 (Ovo Like Transcriptional Repressors 1 and 2), had previously been suspected to inhibit epithelial–mesenchymal transition in cancer cells [54,55] with a hierarchically dominant position for GRHL2, which directly induces expression of EHF and OVOL2 [56,57] (Fig 5B). Moreover, GRHL2 is considered a master regulator of epithelial cell differentiation [58,59]. In line with our previous characterization of genes from C3, *Grhl2* displays H3K27me3-mediated repression in hepatocytes (Fig 5C). We next set out to experimentally define if GRHL2 could induce expression of genes reminiscent of those activated upon HCT in bipotential mouse embryonic liver (BMEL) cells, which have the ability to differentiate both in hepatocytes and cholangiocytes [60,61]. Ectopic expression of GRHL2 (Figs 5D and S7A Fig) triggered induction of a cholangiocyte-like transcriptional program. Indeed, RNA-seq data mining using biological pathway enrichment analyses indicated that expressing GRHL2 in BMEL cells increased the expression of genes related to cell fate commitment including epithelial cell differentiation (Figs 5E and S7B and S5–S6 Tables). In line, GSEA pointed to stimulation of expression of the Epith-Signature and Chol-enriched genes (Fig 5F). Accordingly, when BMEL were grown in three-dimensional Matrigel, GRHL2 promoted acquisition of cell aggregates with a cystic organization (Figs 5G, 5H, and S8) typically adopted by cholangiocytes [62,63].

To more directly assess the ability of GRHL2 to promote HCT, we performed similar experiments in the HepG2 hepatic cell line (Figs 6A and S7C). Although the global reprogramming of the transcriptome was less pronounced than in BMEL cells (S7B and S7D Fig and S7 and S8 Tables), GRHL2 again promoted expression of an archetypal epithelial transcriptional program as judged both through biological pathway enrichment analyses (Fig 6B) and GSEA using the Epith-Signature and Chol-enriched gene sets (Figs 6C and S7E). This was not associated with a decrease in hepatocyte-enriched gene expression (S7F Fig and S2 Table). To further characterize how GRHL2 induces transcriptional reprogramming, we defined its cistrome using chromatin immunoprecipitation (ChIP)-seq in HepG2 cells. Importantly, the GRHL2 binding sites we identified in HepG2 cells (top 10,000 peaks) also displayed recruitment of endogenous GRHL2 previously defined in primary human bronchial epithelial (HBE) cells [64] or an ovarian epithelial cancer cell line [65–67] (S4 Table), albeit with lower enrichment (Fig 6D). This indicated that, while forced TF expression can lead to spurious chromatin binding and gene regulation, chromatin binding of ectopically expressed GRHL2 in hepatocytes recapitulates that of the endogenously expressed protein [64,68]. Moreover, the CistromeGO tool [69] indicated that GRHL2 peaks from HepG2 cells were enriched near genes related to terms consistent with transcriptional remodeling towards an archetypal epithelial cell program such as cell adhesion and junction (Fig 6E). We next inspected GRHL2 ChIP-seq data at genes selected on the basis of their induction upon HCT and GRHL2 ectopic expression as well as their presence in the Epith-Signature and/or cholangiocyte-enriched gene list. In addition to the previously identified GRHL2 target genes *EHF, CLDN4* (*Claudin 4*) and *RAB25* (*RAB25, Member RAS Oncogene Family*) [56,62], these analyses also pointed to direct regulation of genes such as *ESRP1* (*Epithelial splicing regulatory protein 1*), *ITGA3* (*Integrin Subunit Alpha 3*), *ITGB8* (*Integrin Subunit Beta 8*), *JUP* (*Junction* Plakoglobin), and *KRT7* (Fig 6F and 6G and S9 Table). These genes have a proven role in establishment and maintenance of epithelial cell identity and epithelium integrity [62,70–74]. The sensitivity of these genes to GRHL2-mediated up-regulation was further supported by their induction upon moderate ectopic expression of GRHL2 (S9 Fig). As an additional model of human hepatocytes, we used differentiated HepaRG cells [75,76]. GRHL2 overexpression in these cells (Fig 6H) also triggered induction of *CLDN4, EHF, JUP, KRT7*, and *RAB25* (Fig 6I). We also induced GRHL2 expression in human organoids issued from EPCAM-positive cells expanded

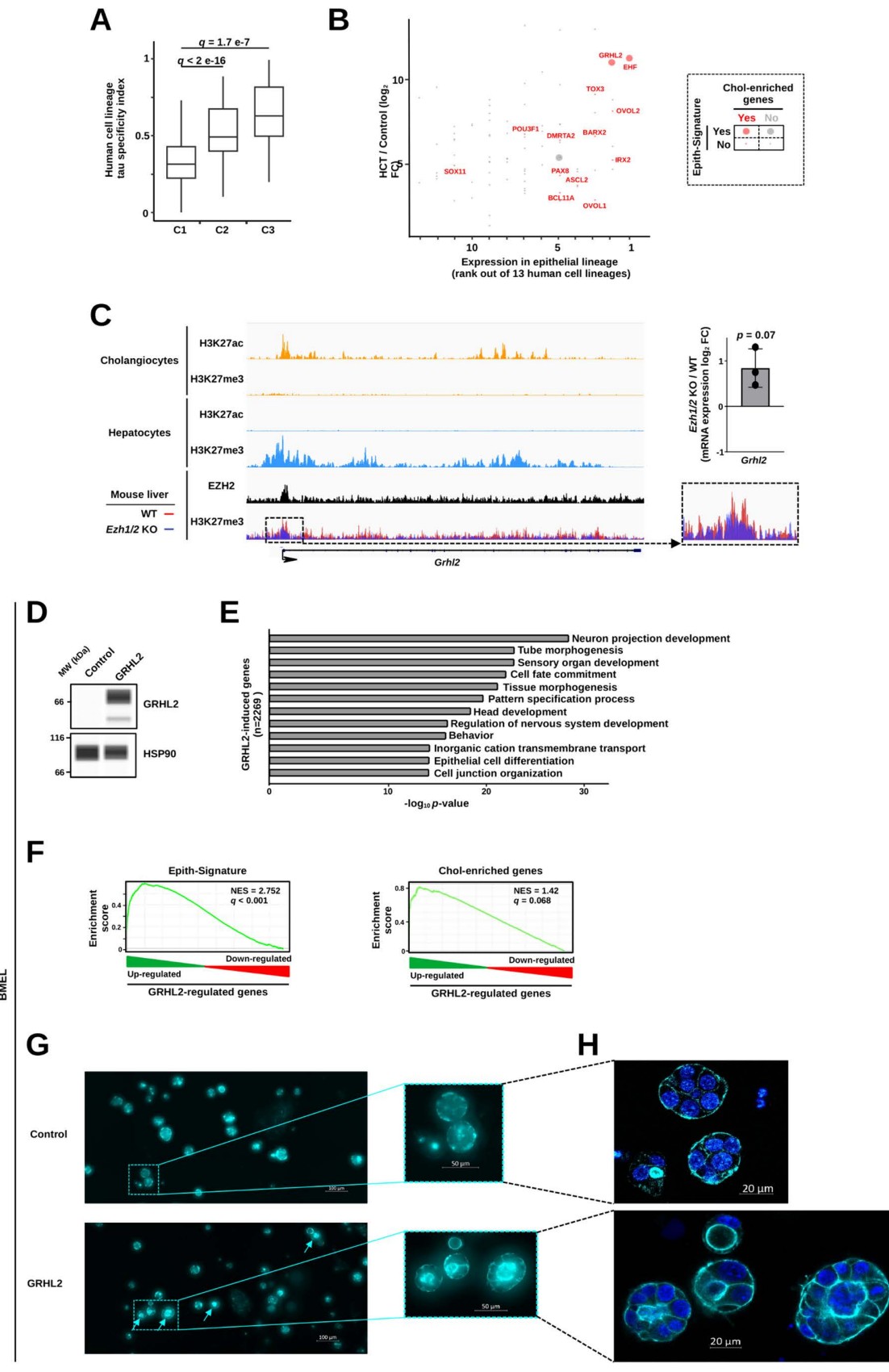

**Fig 5. HCT involves induction of the master epithelial lineage TF GRHL2. (A)** Box plot showing the tau specificity indexes of TF-encoding genes from clusters C1–3 defined in Fig 3. Thirteen human cell lineages (epithelial cells and adipocytes, blood and immune cells, endocrine cells, endothelial cells, germinal cells, glial cells, mesenchymal cells, muscle cells, neuronal cells, pigment cells, tropoblast cells and undifferentiated cells) were used to compute the tau specificity index using the tspex tool [136]. Gene expression in a given cell lineage was defined as the median expression of individual cell types comprised in that lineage. Statistical significance was assessed using a Kruskal–Wallis test with Dunn's multiple comparison post hoc test. **(B)** Scatter plot of individual TF-encoding genes from cluster C3 (as defined in Fig 3) showing their expression in the epithelial lineage (based on data from panel A; lineages were ranked from highest to lowest expression levels for each gene) and their induction levels upon HCT (similar to those used in Fig 3). Additionally, presence of individual TF-encoding genes within the Epith-Signature was highlighted using large dots while identification as a cholangiocyte-enriched gene among liver cell types was highlighted using red dots. **(C)** Integrated Genomics Viewer (IGV) [141] was used to visualize the indicated ChIP-seq profiles from mouse primary cholangiocytes, hepatocytes or liver at the *Grhl2* gene. H3K27ac and H3K27me3 ChIP-seq data from cholangiocytes and hepatcoytes were processed similarly and visualized on the same scale in the two cell types. H3K27me3 ChIP-seq signals in the mouse liver of WT and *Ezh1/2* KO mice were overlayed and a magnification of the *Grhl2* promoter area is shown on the right together with a bar graph showing changes in *Grhl2* expression in the liver these animals computed from RNA-seq data [47]. Mean $\pm$SD together with individual biological replicates are shown. Two-sided one-sample *t* test was used to determine if the mean of log$_2$ FC was statistically different from 0. **(D)** Immunoblotting performed using the Wes system to monitor GRHL2 and HSP90 levels in BMEL cells transfected with a GRHL2 expression plasmid or an empty control construct. Data are representative of those obtained in 3 independents biological replicates. MW, molecular weight. **(E)** Bar plot showing biological term enrichments (top 12) within genes upregulated upon GRHL2 ectopic expression in BMEL cells, i.e., genes with log$_2$ FC > 0 and *q*-value < 0.05 as defined through RNA-seq analyses on 4 independent biological replicates. Data were obtained using Metascape [126]. **(F)** Enrichment plots obtained using GSEA performed with the Epith-Signature [36] or Chol-enriched genes [34] as the gene sets and transcriptomic changes induced by GRHL2 ectopic expression in BMEL cells (RNA-seq data from 4 independent biological replicates used in panel E). NES stands for normalized enrichment score. **(G)** BMEL cells were transfected with a GRHL2 expression plasmid or an empty control construct and grown in Matrigel. Representative images of obtained aggregates stained with phalloidin are shown. The arrows point to cystic cell aggregates. **(H)** Cell aggregates highlighted in the zoomed areas from panel G were further analyzed using confocal microscopy in order to better display the differences in their organization highlighting the cyst structures adopted by BMEL cells overexpressing GRHL2. Images were acquired using LSM confocal microscopy. To smooth the images, LSM image processing was applied, and due to the low signal in control images, a median filter was also used to improve image quality. The original data underlying this figure can be found at the Gene Expression Omnibus (GSE281717).

in three-dimensional Matrigel and induced to differentiate according to [77]. We found that this procedure gives rise to a population of cells expressing both hepatocyte and cholangiocyte markers [78]. GRHL2 triggered moderate induction of several of its target genes concomitant with reduced expression of the *ALB* and *CYP3A4* (*Cytochrome P450 family 3 subfamily A member 4*) hepatocytes markers (Fig 6J).

Finally, the impact of forced GRHL2 expression in mouse hepatocytes in vivo was assessed following hydrodynamic tail vein injection of the GRHL2-encoding plasmid (Figs 7A and S10A) or injection of recombinant adeno-associated viruses (AAV8) (Figs 7B and S10B). Reminiscent of our in vitro findings, we observed that GRHL2 induces expression of the cholangiocyte-enriched epithelial genes *Cldn4*, *Ehf*, *Esrp1*, *Itga3*, *Itgb8*, *Jup*, *Krt7,* and *Rab25* when expressed in hepatocytes in vivo (Fig 7C and 7D). Concomitantly, expression of hepatocyte marker genes such as *Alb*, *Gsta3* (*Glutathione S-transferase, alpha 3*) or *Fmo5* (*Flavin containing monooxygenase 5*) was reduced (Fig 7E). Expression of the hepatic bile acid exporter *Abcb11* (*ATP-binding cassette, sub-family B member 11*; *Bsep*) was also diminished while that of *Cyp3a11* was increased, changes characterizing mice with altered hepatic bile acid metabolism [79,80]. While the liver architecture was not affected after 1 week (S10C Fig), dramatic histological alterations to the hepatocyte parenchyma, characterized by gained heterogeneity in cell shape and size (Figs 7F and S10C), and to immune cell infiltration (S10D Fig) was observed after 2 weeks.

Taken together, these results indicate that GRHL2 promotes direct HCT through activation of genes characteristic of an archetypal epithelial cell program.

## Induction of GRHL2 in hepatocytes undergoing HCT in human livers

We and others have reported that livers of patients with end-stage chronic liver diseases are characterized by populations of hepatocytes displaying signs of HCT, i.e., concomitant loss of hepatocyte identity gene expression and induction of cholangiocyte markers [11,13–15,81,82]. To assess GRHL2 expression and its link to HCT in human livers, we first used samples from patients with ALD-related decompensated cirrhosis. Fragments of healthy liver

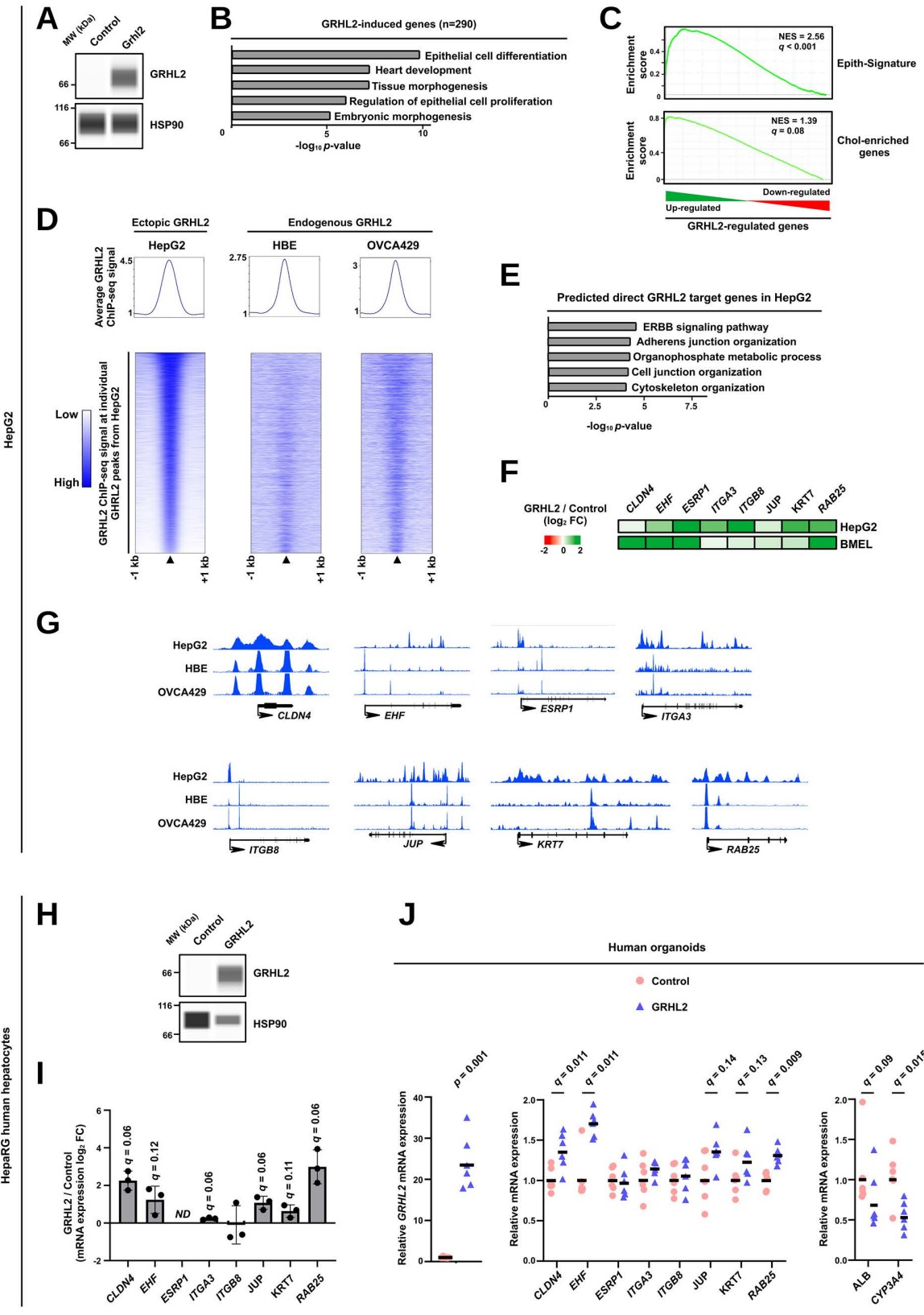

**Fig 6. GRHL2 ectopic expression in hepatocytes triggers induction of epithelial genes involved in HCT. (A)** Immunoblotting performed using the Wes system to monitor GRHL2 and HSP90 levels in HepG2 cells transfected with a GRHL2 expression plasmid or an empty control construct. Data are representative of those obtained in 3 independents biological replicates. MW, molecular weight. **(B)** Bar plot showing biological term enrichments (top 5) within genes upregulated upon GRHL2 ectopic expression in HepG2 cells, i.e., genes with $\log_2 FC > 0$ and $q$-value $< 0.05$ as defined through RNA-seq analyses on 4 independent biological replicates. Data were obtained using Metascape [126]. **(C)** Enrichment plot obtained using GSEA performed with the Epith-Signature [36] or Chol-enriched genes [34] as the gene sets and transcriptomic changes induced by GRHL2 ectopic expression in HepG2 cells (RNA-seq data from 4 independent biological replicates used in panel B). NES stands for normalized enrichment score. **(D)** The top 10,000 GRHL2 binding sites identified in HepG2 cells were used to monitor GRHL2 ChIP-seq signal from the indicated cells. Human bronchial epithelial (HBE) are primary human bronchial epithelial cells while OVCA429 is an ovarian epithelial cancer cell line. Both average ChIP-seq signal (top; background signal was arbitrarily set to 1) and signal at individual binding sites (heatmap at the bottom) are shown in 2 kb regions centered on GRHL2 peaks. Individual binding sites were ranked according to the GRHL2 ChIPseq signal from HepG2 cells. GRHL2 binding sites and ChIP-seq signal in HepG2 cells were defined from 4 independent biological replicates (see Materials and methods). **(E)** Bar plot showing biological term enrichments (top 5) within genes predicted to be directly targeted by GRHL2 in HepG2 cells by the CistromeGO tool [69] on the basis of identified GRHL2 binding sites. **(F)** Heatmap showing changes in expression of the indicated genes in the RNA-seq data obtained from HepG2 or BMEL (Fig 5) cells ectopically expressing GRHL2. **(G)** Integrated Genome Browser (IGB) [140] was used to visualize GRHL2 ChIP-seq profiles from HepG2, HBE and OVCA429 (same datasets as those used in panel D) at the indicated target genes (*CLDN4*, *EHF*, *ESRP1*, *ITGA3*, *ITGB8*, *JUP*, *KRT7,* and *RAB25*). **(H)** Immunoblotting performed using the Wes system to monitor GRHL2 and HSP90 levels in HepaRG cells differentiated into hepatocytes and electroporated with a GRHL2 expression plasmid or an empty control construct. Data are representative of those obtained in 4 independents biological replicates. MW, molecular weight. **(I)** RT-qPCR data showing the expression of the indicated GRHL2 target genes in HepaRG cells differentiated into hepatocytes and electroporated with a GRHL2 expression plasmid or an empty control construct ($n = 3$). Log$_2$ FC in gene expression in cells expressing GRHL2 compared to control cells (transfected with an empty plasmid) are shown in the bar graph, which displays means $\pm$SD together with individual biological replicates. Two-sided one-sample $t$ test with Benjamini–Hochberg correction was used to determine if the mean of $\log_2$ FC was statistically different from 0. ND stands for not detected. The numerical values of all biological replicates can be found in the S1 Data file. **(J)** RT-qPCR data showing the relative mRNA expression of *GRHL2* (left), its target genes (middle) or hepatocyte markers (right) in human organoids transduced with GRHL2 or GFP encoding AAV3 ($n = 6$). Individual data together with the geometric means (horizontal bars) are shown. Fold changes were obtained using the average of the control group, arbitrarily set to 1 for each analyzed gene, as the reference. One-sided Mann–Whitney $U$ test with Benjamini–Hochberg correction for multiple testing was performed to determine whether gene expression was significantly induced upon GRHL2 ectopic expression. The numerical values of all biological replicates can be found in the S1 Data file. The original data underlying this figure can be found at the Gene Expression Omnibus (GSE281717).

samples from patients subjected to liver resection due to hepatic tumors were used as control in these experiments. First, we found that GRHL2 expression was induced in patients with liver failure both at the mRNA and protein levels (Fig 8A and 8B). This was associated with decreased EZH2 and H3K27me3 levels in those livers (Fig 8B). To define that GRHL2 was specifically induced in hepatocytes, we monitored its mRNA levels in RNA-seq-based transcriptomic data previously obtained using microdissection of hepatocyte-enriched areas from those livers [14]. Compared to control samples, GRHL2 expression was induced in ALD-related liver failure ($\log_2$ FC = 1.38, $q$-value = 0.11), which correlated with the induction of genes defined as up-regulated during HCT as well as genes from the Epith-Signature (Fig 8C). Moreover, unlike control tissues which stained positive for GRHL2 in cholangiocytes only, GRHL2 immunostaining in livers from ALD patients was positive not only in neoductules but also in hepatocytes found at the border of the regenerative nodules next to the fibrotic bands (Figs 8D, S11A, and S11B). Immunofluorescence co-staining of GRHL2 and albumin (ALB) showed a gradual increase in GRHL2 together with a gradual loss of ALB staining in hepatocytes bordering the fibrotic scar (S11C Fig). Additional co-staining for KRT7 further revealed triple-positive cells at the scar interface (Figs 8E, S11D, and S11E), in line with HCT occurring in this specific micro-environment in human cirrhotic livers [83]. Single-nuclei RNA-seq data analysis of human livers at different stages of MASLD [11] showed increased GRHL2-expressing hepatocytes in end-stages of the disease (i.e., cirrhosis and liver failure) when compared to healthy control livers (S12A–S12D Fig). Moreover, GRHL2 expression was found in hepatocytes acquiring mixed expression of hepatocyte and cholangiocyte identity genes consistent with its HCT-promoting activities (S12E Fig). Finally, GRHL2 immunostaining in samples from patients with liver failure due to primary biliary cholangitis (PBC) or PSC also revealed GRHL2-positive hepatocytes (Figs 8F and S13).

Overall, these data indicate that induction of GRHL2 characterizes HCT in human livers.

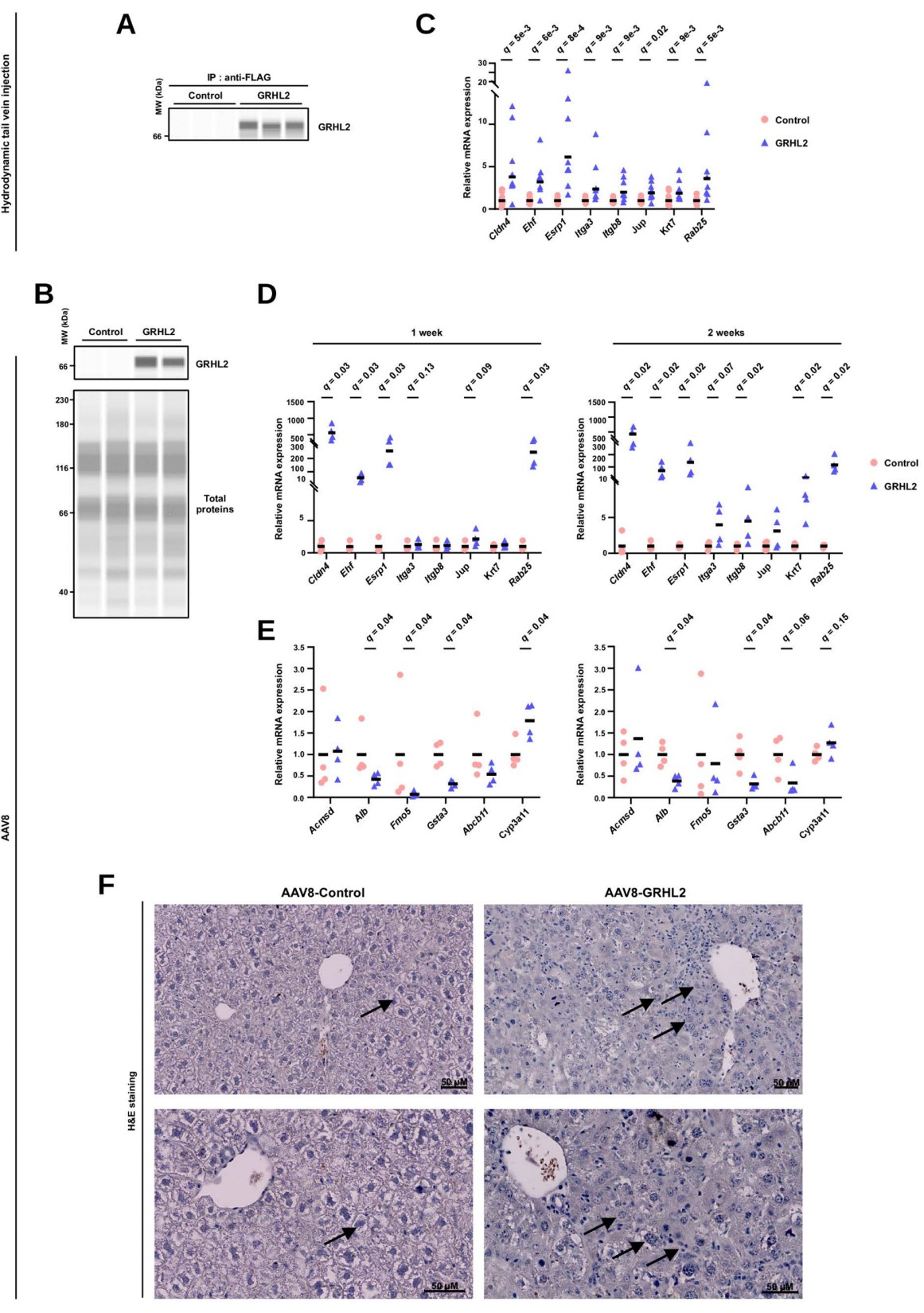

**Fig 7. GRHL2 ectopic expression triggers HCT in vivo in the mouse liver.** **(A)** Immunoblotting performed using the Wes system to monitor GRHL2 levels in the liver of mice subjected to hydrodynamic injection with the flag-tagged GRHL2-encoding plasmid or an empty construct (control). Nuclear extracts from individual mouse livers (3 mice per experimental group) were subjected to immunoprecipitation with an antibody against flag and immunoblotted using an anti-GRHL2 antibody. S10 Fig shows the immunoprecipitation specificity control. MW, molecular weight. **(B)** Immunoblotting performed using the Wes system to monitor GRHL2 levels in the liver of mice injected with GRHL2 or GFP (control) encoding AAV8. Nuclear extracts from individual mouse livers (2 mice per experimental group) were prepared after 1 week and subjected to immunoblotting using an anti-GRHL2 antibody. A total protein assay was used for loading control. MW, molecular weight. **(C)** RT-qPCR data showing the relative mRNA expression of the indicated GRHL2 target genes in the liver of mice subjected to hydrodynamic injection with the flag-tagged GRHL2-encoding plasmid ($n = 8$ mice) or an empty construct (control; $n = 9$ mice). For each gene, data for individual mice are shown together with the geometric mean (horizontal bar). Fold changes were obtained using the average of the control group, arbitrarily set to 1 for each analyzed gene, as the reference. One-sided Mann–Whitney $U$ test with Benjamini–Hochberg correction for multiple testing was performed to determine whether gene expression was significantly induced upon GRHL2 ectopic expression. The numerical values of all individual mice can be found in the S1 Data file. **(D)** RT-qPCR data showing the relative mRNA expression of the indicated GRHL2 target genes in the liver of mice 1 or 2 weeks after being injected with GRHL2 ($n = 4$ mice) or GFP (control; $n = 4$ mice) encoding AAV8. For each gene, data for individual mice are shown together with the geometric mean (horizontal bar). Fold changes were obtained using the average of the control group, arbitrarily set to 1 for each analyzed gene, as the reference. One-sided Mann–Whitney $U$ test with Benjamini–Hochberg correction for multiple testing was performed to determine whether gene expression was significantly induced upon GRHL2 ectopic expression. The numerical values of all individual mice can be found in the S1 Data file. **(E)** RT-qPCR data showing the relative mRNA expression of the indicated hepatocyte marker genes in the liver of mice 1 or 2 weeks after being injected with GRHL2 ($n = 4$ mice) or GFP (control; $n = 4$ mice) encoding AAV8. Data were plotted and analyzed as in panel D. The numerical values of all individual mice can be found in the S1 Data file. **(F)** Representative images obtained using liver slices from mice injected with GRHL2 or GFP (control) encoding AAV8 stained with hematoxylin and eosin (H&E) after 2 weeks. The arrows show the archetypal hepatocyte structure in the control livers which contrasts with the heterogenous cell shapes observed in GRHL2 overexpressing livers. Additional images are shown in S10C Fig.

## Discussion

Hepatocyte plasticity is of primary importance for the regenerative potential of the liver. As such HCT assists the liver when coping with compromised or exhausted renewal potential of cholangiocytes [12,84]. Thus, understanding the molecular basis of HCT is of primary interest for regenerative medicine in the field of bile duct diseases including ALGS and intrahepatic biliary injuries such as PBC and PSC [85]. Beyond such conditions, HCT has also been described as a feature of end-stage disease-associated hepatocytes in ALD and MASLD [11,14,15]. In this specific context, whether HCT is indicative of ongoing regeneration in cirrhotic livers or is a phenomenon contributing to precipitate liver failure remains to be established.

Changes to the hepatocyte microenvironment are main drivers of HCT. For instance, cytokines such as transforming growth factor beta are instrumental in inducing this process [12,86,87], which occurs in human ALD-related cirrhosis at the scar interface (this study and [83]). Here, we show that HCT is linked to expression of GRHL2 in hepatocytes lining the fibrotic bands in livers of patients with end-stage ALD, PBC, and PSC. This implies that loss of hepatocyte intercellular junctions and interaction with stiff extracellular matrices likely also contribute to HCT. Indeed, this can lead to activation of the Hippo/YAP signaling pathway, a well-established driver of cholangiocyte differentiation and bile duct development [14,88,89]. Moreover, alterations in the unique multi-polarization of hepatocytes could also contribute to reprogram their transcriptomes. Indeed, modulation of the cellular polarity during differentiation processes is not merely a consequence, but rather actively contributes to cell fate decisions [90,91]. Hence, HCT is most probably triggered by a complex interplay of signals. Defining how these different mechanisms operate in concert to drive HCT and, in particular, how they are functionally connected with the activities of TFs such as GRHL2 will further improve our understanding of this process.

Previous reports have posited that hepatocytes acquire a progenitor-like hepatobiliary identity in injured livers [84]. However, this assumption has usually been based on the co-expression of a limited number of hepatocyte- and cholangiocyte-specific markers including notably SOX9 [18,19]. However, careful examination of data reported in [17] indicates that only a limited subset of the genes induced in so-called "progenitor-like cells" have stronger expression levels in hepatoblasts than in hepatocytes. Moreover, this study also pointed to gene induction in these cells involving transcriptional enhancers different from the ones active in hepatoblasts. Although relying on data issued from hepatocyte lineage

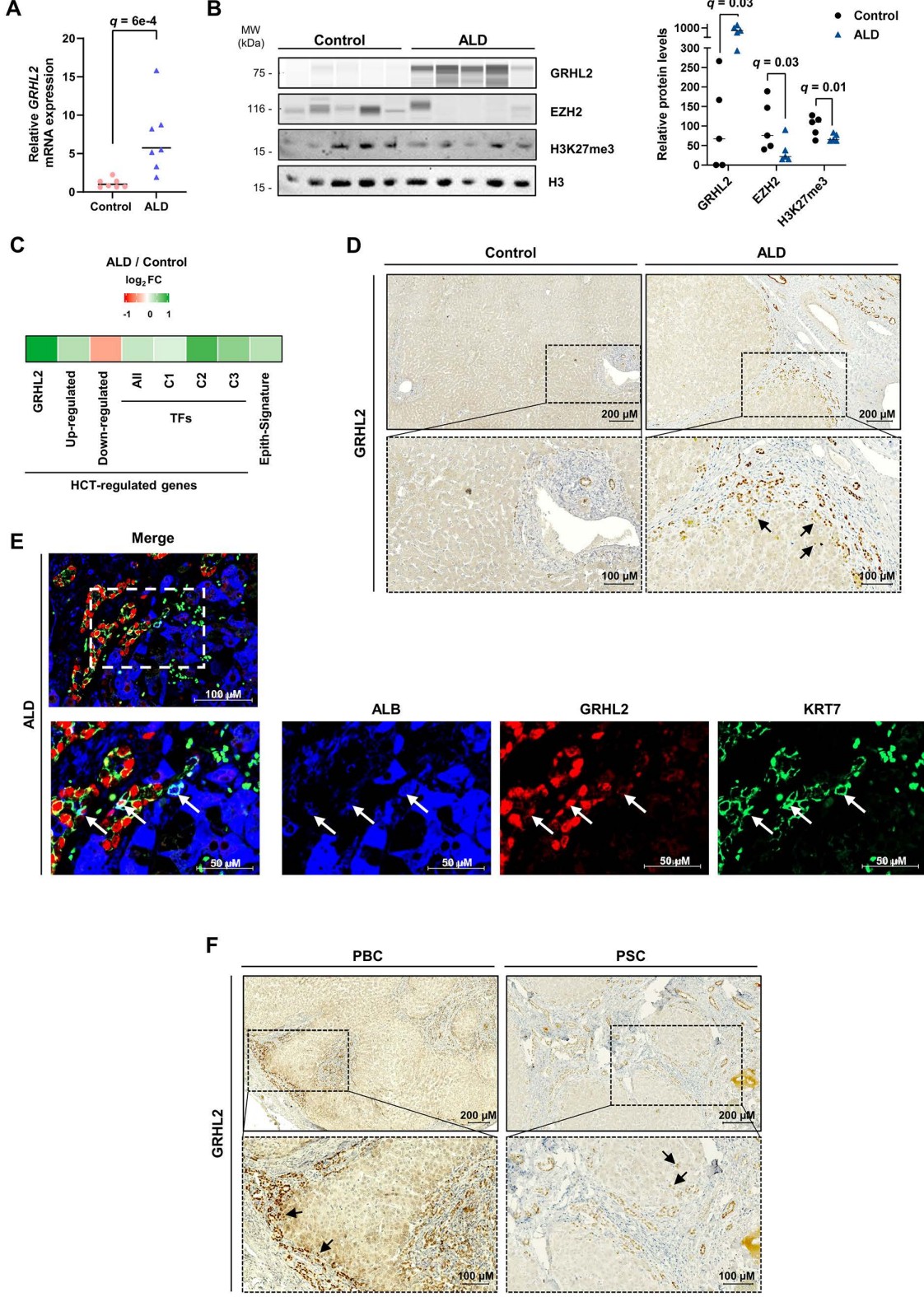

**Fig 8. Hepatocytes in end-stage human chronic liver diseases express GRHL2 and epithelial genes associated with HCT. (A)** RT-qPCR data showing relative *GRHL2* mRNA expression in human samples from control and ALD-related liver failure (*n* = 7). Data from individual samples are shown

together with the geometric mean (horizontal bar). Fold changes were obtained using the average of the control group, arbitrarily set to 1. One-sided Mann–Whitney *U* test was performed to determine whether GRHL2 expression is significantly higher in samples from the liver failure group. The numerical values of all individual samples can be found in the S1 Data file. **(B)** Immunoblotting performed using the Wes system to monitor GRHL2, EZH2, H3K27me3, and H3 levels in human samples from control and ALD-related liver failure (*n* = 5). The bar graph on the right shows results of signal quantifications normalized to H3. One-tailed Mann–Whitney *U* test with Benjamini–Hochberg correction for multiple testing was used to define whether protein levels were significantly higher in samples from the liver failure group. MW, molecular weight. The numerical values of all individual samples can be found in the S1 Data file. **(C)** RNA-seq analyses using microdissected hepatocyte-enriched areas from control and ALD-related human liver failure from our previous study [14] (*n* = 5 and 7 for the control and liver failure groups, respectively) were used to monitor differential expression of *GRHL2* or the indicated previously defined groups of genes (median log2 FC is reported). **(D)** Representative immunostaining of GRHL2 in human livers from the control and ALD-related liver failure groups. Results obtained with liver samples from 5 patients are provided in S11A and S11B Fig Zoomed images of the area delimited by the dotted rectangles are shown at the bottom. Arrows point to examples of GRHL2-positive hepatocytes. **(E)** Co-immunostaining of the hepatocyte marker Albumin (ALB; in blue), GRHL2 (in red), and the cholangiocyte marker KRT7 (in green) in ALD-related human liver failure (Patient ALD#1 also used in panel D and in S11C–S11E Fig). Zoomed images of the area delimited by the dotted rectangle in the top image are shown at the bottom. Arrows indicate triple positive cells for ALB, GRHL2, and KRT7. Additional fields are shown in S11C–S11E Fig. **(F)** Representative immunostaining of GRHL2 in human livers from patients with PBC- or PSC-related liver failure. Results obtained with liver samples from 3 patients with PBC or PSC are provided in S13 Fig, respectively. Zoomed images of the area delimited by the dotted rectangles are shown at the bottom. Arrows point to examples of GRHL2-positive hepatocytes.

tracing is not possible in humans, scRNA-seq analysis of livers from patients with MASLD-associated liver failure pointed to HCT occurring in the absence of cells displaying a progenitor-like state [11]. These findings are consistent with our study, which also indicates that ectopic GRHL2 expression is sufficient to directly induce HCT genes in mature hepatocytes. Altogether, these data establish that HCT is a direct conversion process not requiring reversion to a progenitor-like cellular state.

Our study points to induction of a transcriptional program characterizing archetypal epithelial cells with a single apico-basal axis as the primary gene expression reprogramming event underlying HCT. In this context, GRHL2 induces epithelial genes such as genes involved in tight junctions and promotes acquisition of cyst-like structures. GRHL2 induced expression of some classical cholangiocyte markers (e.g., *KRT7*), but not all commonly used ones, indicating GRHL2 is most likely not sufficient to drive full HCT. GRHL2's role in promoting HCT is most probably incorporated into a network of collaborating TFs. For instance, GRHL2 promoted expression of the epithelial transcriptional regulators *EHF* and *ESRP1* [92,93]. Accordingly, spatial transcriptomics in human cirrhotic livers indicated that areas where HCT occurs are enriched for expression of GRHL2 and several of its target genes we identified in our study including *ESRP1* [83]. Besides forming the bile ducts, a subset of cholangiocytes is also equipped with the ability to modify the hepatocyte-secreted bile and control its flow [94]. This involves genes such as *Cystic Fibrosis Transmembrane Conductance Regulator* (*CFTR*), which were not significantly induced by GRHL2 in our experimental models. Additional regulatory pathways such as those induced by retinoic acid may be required to further modulate cholangiocyte functionalities [95]. Besides *GRHL2*, we also confirmed the previously described induction of *SOX4* (*SRY-Box Transcription Factor 4*) (S3 Table). In addition to promoting HCT by operating as a pioneer factor, SOX4 also directly represses the hepatocyte transcriptional program [96]. We also observed reduced expression of hepatocyte markers upon GRHL2 ectopic expression in vivo in mouse hepatocytes together with profound alteration of the hepatic parenchyma structure. These data suggest that GRHL2 contributes to the repression of hepatocyte identity although this warrants further investigation. Finally, GHRL2-induced changes in gene expression in HCT may also result from the regulation of non-coding RNAs such as *MIR122* [97,98].

Induction of the cholangiocyte transcriptional program involves epigenetic remodeling events including removal of H3K27me3. A recent assessment of the epigenetic control of HCT identified a role for enzymes controlling H3K36me3 (NSD1 and KDM2A) but not for EZH and H3K27me3 [99]. However, *Ezh1* and *Ezh2* were silenced individually, which most probably explains these results since these two enzymes are redundant in hepatocytes and combined deletion of the two genes is required to trigger loss of H3K27me3 [100]. In ALD-related liver failure samples, we found decreased EZH2 and H3K27me3 levels, whose functional consequences is supported by a recent study showing that even a transient PRC2 inhibition is sufficient to trigger irreversible induction of TF encoding genes [101]. A role for H3K27me3 in controlling

hepatoblast-to-cholangiocyte differentiation has been reported in a study showing that EZH2 inhibition promotes cholangiocyte differentiation [102]. This is in line with our findings that loss of EZH2-mediated repression is involved in HCT even if HCT does not per se phenocopy the developmental process. Hence, a better understanding of the precise kinetics of epigenetic remodeling events and their functional connection to TF activities along the course of HCT is warranted.

## Materials and methods

### Ethics statement

Work performed in this study using human samples is authorized by the Lille ethical committee (Lille University Hospital), and informed consent was obtained from all subjects. The research was carried out according to the World Medical Association Declaration of Helsinki. All animal studies were performed in compliance with EU specifications regarding the use of laboratory animals and have been approved by the Nord-Pas de Calais Ethical Committee (CEEA-075; approval numbers 44535-2023030722295795 v6 and 30418-20210824091242 v8).

### Cell culture, transfection, and treatments

HepG2 cells were cultured in William's E medium (Gibco, 22551022) supplemented with 10% FBS (Life, SVF30160.03), 20 mU/mL bovine insulin (Sigma, I5550), 50 mM dexamethasone (Sigma, D1756), and 100 U/mL penicillin-streptomycin (Gibco, 15140). Plates were coated with gelatin 1 g/L (Sigma, G1890). BMEL cells, described in [60], were cultured in RPMI 1640 Glutamax (Gibco, 61870-044), supplemented with 10% FBS (EuroBio, CVFSVF00-01), 30 mg/mL IGFII (PeproTech, 100-12-100 µg), 50 ng/mL mouse EGF (PeproTech, 315-09-1 mg), 10 µg/mL bovine insulin, and 100 U/mL penicillin-streptomycin. BMEL cells were grown on type I collagen-coated plates. All cells were grown at 37°C in a humidified atmosphere with 5% $CO_2$.

HepaRG cells previously described in [75] were seeded at the density of $2.6 \times 10^4$ cells/cm$^2$ and maintained in a Williams E medium supplemented with 10% FCS, 100 units/mL penicillin, 100 µg/ml streptomycin, 5 µg/mL insulin, and $5 \times 10^{-5}$ M hydrocortisone hemisuccinate. After 15 days, cells were maintained in the same medium supplemented with 2% dimethyl sulfoxide (DMSO) for 2 more weeks to obtain complete hepatocyte differentiation. Differentiated cells were then detached with trypsin and resuspended in Williams E medium without DMSO prior to electroporation.

RT-qPCR analyses indicated that *GRHL2* expression was low or not detectable in all these cell lines when grown in basal conditions.

Pharmacological inhibition of EZH1 and EZH2 was obtained by treating cells with a combination of 1µM of UNC1999 (Target Mol, T3057) [103] and 1µM of Valemetostat Tosylate (MedChem Express, 1809336-93-3) [104] for 4 days. DMSO 0.02% was used as control.

Plasmids were transfected either in HepG2 cells or BMEL cells using JetPrime (Polyplus transfection) according to the manufacturer's instruction. The GRHL2 encoding construct was purchased from VectorBuilder. It allowed for expression of the GRHL2 cDNA (NM_024915.4) fused to a flag tag under the control of an *EEF1A1* (*EF1A*) promoter. This bicistronic construct also allowed for expression of enhanced green fluorescent protein, which was leveraged to monitor transfection efficiency using green fluorescence microscopy. The pcDNA3 vector (Invitrogen) was used as control. Cell culture medium was refreshed after 24 h and cells were harvested 48 h after transfection.

Plasmids were electroporated in differentiated HepaRG cells using the Neon electroporation system (Thermo Fisher Scientific) as previously described [105]. Briefly, $10^6$ HepaRG cells were electroporated with 2 µg of control or GRHL2 encoding plasmids with a single pulse of 1,500 volts for 20 ms. The cells were then plated at low density ($2.6 \times 10^4$ cells/cm$^2$) in Williams E medium without DMSO. After 24 hours, the medium was renewed. Forty-eight hours after electroporation, the cells were harvested in ice-cold phosphate-buffered saline (PBS), scraped and pelleted by centrifugation at 400 g during 5 min.

## Three-dimensional cell culture

BMEL cells transfected using JetPrime as detailed hereabove were 48 h latter used for three-dimensional culture in Matrigel (Corning, 356255). 20,000 cells were resuspended in 5 µL of culture medium and gently mixed with 30 µL of Matrigel using a pre-chilled pipette tip, taking care to avoid introducing air bubbles. Matrigel-embedded cells were deposited at the center of pre-warmed Ibidi plates (IBIDI, 80806). Plates were incubated at 37°C for 20 min to allow the Matrigel to solidify before adding complete medium to each well.

## RNA extraction and gene expression analyses using real-time PCR

Tissues were homogenized using Minilys and 1.4 mm Ceramic beads (Bertin Technology) (3 × 15 s, 5,000 rpm). Total RNA was extracted from cell lines and tissues using the Nucleospin RNA kit (Macherey-Nagel, 740955) following the manufacturer's protocol. RNA was reverse-transcribed using high-capacity cDNA Reverse Transcription Kit (Applied Biosystem). Quantitative PCR (qPCR) was performed on a QuantStudio 3 (Applied Biosystem) with the Fast program (Applied Biosystem) using the PowerTrack SYBR Green Master Mix (Applied Biosystem). The specificity of amplification was checked by recording the dissociation curves, and the PCR efficiency was verified to be above 95% for each primer pair. mRNA levels were normalized to the average expression of housekeeping genes (*TBP* only for human livers and organoids) and the fold induction was calculated using the ΔΔCT method. The sequences of primers used are listed in S10 Table. To measure *GRHL2* and *EHF* mRNA expression levels, duplex real-time PCR using the TaqMan technology (Applied Biosystem) was performed. FAM (6-carboxyfloresceine)-labeled probes against *GRHL2* (ThermoFisher, Hs00227745) and VIC (2′-chloro-7′phenyl-1,4-dichloro-6-carboxy-fluorescein)-labeled probes against ribosome RNA 18S (housekeeping control) were used together with the TaqMan Fast Advanced Master Mix (Applied Biosystem) according to the manufacturer's instructions.

Alternatively, RNA integrity and quantity were evaluated using the Agilent 2100 Bioanalyser (Agilent Technologies) before samples were used for library preparation and high-throughput sequencing.

## Protein extraction

**Total cellular extracts.** For total protein extraction, cells were washed twice with ice-cold PBS, scraped and pelleted by centrifugation at 400 g during 5 min. Cell pellets were resuspended in high salt lysis buffer [25 mM Tris-HCl pH 7.5, 500 mM NaCl, 1 mM EDTA, 0.5% NP40, 0.9 mM Na Butyrate, 0.1 mM NA$^+$ Orthovanadate (Sigma, 450243), deacetylase inhibitor cocktail (MCE; HY-K0030), 83 U/µL Benzonase (Sigma, E1014-25KU) and protease inhibitor cocktail (PIC) (Roche; 63592800)]. After 10 min of incubation at 4°C, samples were sonicated twice for 10 min (10 cycles of 30 s ON/30 s OFF) with Bioruptor Pico (Diagenode). Samples were next centrifugated at 16,000$g$ for 5 min at 4°C and supernatants were transferred to new tubes. An aliquot was used to determine protein concentration using the Pierce BCA protein assay kit (Thermo scientific) and Laemmli 6× (175 mM Tris-HCL pH 6.8, 15% glycerol, 5% Sodium docetyl sulfate (SDS), 300 mM Dithiothreitol (DTT), and 0.01% Bromophenol blue) was added.

**Chromatin fractionation.** Human liver tissues were homogenized in Buffer A (10 mM HEPES pH 7.5, 10 mM KCl, 1.5 mM MgCl2, 340 mM sucrose, 10% glycerol, 1 mM DTT, PIC) and Triton X-100,0.1% using Polytron (Fisher Scientific, IKA T10 basic ULTRA-TURRAX), followed by 10 min incubation at 4°C. Then, samples were centrifuged at 1,300$g$ for 5 min at 4°C and supernatants were transferred to new tubes subjected to centrifugation at 1,3,000$g$ for 5 min at 4°C in order to remove any insoluble material. Cell pellets were washed with buffer A once and subsequently incubated in solution B (3 mM EDTA, 0.2 mM EGTA, 1 mM DTT, and PIC) for 30 min at 4°C. Samples were centrifuged at 1,700$g$ for 5 min at 4°C and supernatants were discarded. Chromatin pellets were washed a first time with solution B and a second time with PBS, and then resuspended in buffer C (50 mM Tris–HCl pH 8.0, 1 mM MgCl$_2$, and 83 U/µl benzonase). Samples were incubated for 20 min at 4°C before protein quantification and addition of Laemmli buffer 6× as indicated hereabove.

**Nuclear extract and Immunoprecipitation.**  For mouse experiment, the posterior right lobe was systematically used for protein extraction. Liver samples were filtered through 0.7 µm filters in 1 mL of cold PBS and transfered onto 1.5 mL Eppendorf. Samples were centrifugated at 400$g$ at 4°C during 5 min and supernatants were removed. Pellets were suspended into 500 µL of hypotonic buffer (20 mM of Tris-HCl pH 8.0, 10 mM of NaCl, 3 mM of MgCl$_2$, 0.4% NP40, 5 mM of sodium butyrate (Sigma, 303410), PIC, phosphatase, and deacetylase inhibitor cocktails and incubated 5 min at 4°C. Samples were spin at 600$g$ at 4°C during 5 min and supernatants were transfer into new tubes (cytoplasmic fraction). Pellets were suspended in 200 µL of nucleus lysis buffer (25 mM of Tri pH 8.0, 500 mM of NaCl, 1 mM of EDTA, 0.5% of NP40, PIC, 5 mM of sodium butyrate, PIC, phosphatase, and deacetylase inhibitor cocktails. After 20 min of incubation at 4°C, samples were sonicated for 10 min (10 cycles of 30 s ON/30 s OFF) with Bioruptor Pico and then centrifugate at max speed during 5 min at 4°C. Soluble fractions (nuclear extract) were collected and transfer into new tubes. 500 µg of nuclear extract was used per immunoprecipitation and 2 volume of dilution buffer (25 mM of Tris pH 8.0, 500 mM of NaCl, 1 mM of EDTA, 0.5% of NP40, PIC, 5 mM of sodium butyrate, PIC, phosphatase, and deacetylase inhibitor cocktails) was added. Samples were incubated with 2 µg of antibody [anti-IgG and anti-flag (listed in S11 Table)] overnight at 4°C under rotation. Dynal magnetic beads (Sigma, A9418) (5 µL of protein A and 5 µL of protein G per IP) were washed 3 times in PBS-BSA. Beads were block in 500 µL of PBS-BSA (5 mg/mL) overnight at 4°C under rotation before been added into samples. After 4 h of incubation at 4°C, samples were washed 4 times with cold washing buffer (25 mM Tris-HCl pH 8.0, 150 mM NaCl, 1 mM EDTA, 0.2% NP40, PIC, 5 mM of sodium butyrate, PIC, phosphatase, and deacetylase inhibitor cocktails), each time 5 min under rotation at room temperature. Samples were eluted in 40 µL of Laemmli.

## Western blotting

Twenty µg of proteins were separated by 15% SDS-PAGE and immunodetected using primary antibodies (listed in S11 Table). Detection was achieved using HRP-conjugated secondary antibodies (Sigma-Aldrich). Images were acquired with an iBrigth CL1500 Imaging System (Thermo Fisher Scientific).

## Simple western immunoassays (Wes)

Protein extracts were analyzed using a Wes system (Protein Simple) according to the manufacturer's instructions. Samples were diluted in 0.1× sample buffer 2 (Protein Sample). Final protein concentrations used in our assays ranged from 0.4 to 0.5 mg/mL. Separation was performed using the 12–230 kDa capillary cartridges. Primary antibodies used are listed in S11 Table. Secondary antibodies were provided by the manufacturer (PS-MK14 and PS-MK15, Protein Simple). Data were analyzed using the Compass software (Protein Simple). Quantifications were obtained using the area under the peak of the protein of interest. Total protein loading control was performed using the chemiluminescent total protein assay (Protein Simple, DM-TP01).

## Chromatin immunoprecipitation (ChIP)

HepG2 (5 × 10$^6$ cells) were transfected with GRHL2 plasmid using JetPrime (Polyplus transfection) according to the manufacturer's instruction. Cells were collected 48 h after transfection, washed with PBS, and fixed with 1% formaldehyde (Thermo Scientific; 28908) for 10 min at room temperature (RT) before quenching by incubation with 125 mM of glycine for 5 min. After 2 washes with ice-cold PBS, cells were scraped and pelleted by centrifugation at 600 $g$ for 5 min at 4°C. Cell pellets were incubated for 10 min in 0,25% Triton X-100, 10 mM EDTA, 10 mM HEPES, 0.5 mM EGTA, and span 5 min at 600$g$ at 4°C. Supernatants were removed and cell pellets were resuspended and incubated 10 min in 0.2 M NaCl, 1 mM EDTA, 10 mM HEPES, 0.5 mM EGTA, followed by 5 min of centrifugation at 4°C at 600$g$. Nuclei were then resuspended in 600 µL of lysis buffer (50 mM Tris-HCl pH 8.0, 10 mM EDTA, 1% SDS) and sonicated (45 cycles 30 s ON/30 s OFF). All buffers were supplemented with PIC (Roche). Chromatin (30 µg for input, H3K27ac and H3K27me3 ChIP and 200 µg for

GRHL2 ChIP) was diluted 10-fold in dilution buffer (20 mM Tris HCl pH 8.0, 1% Triton X-100, 2 mM EDTA, 150 mM NaCl) and incubated overnight at 4°C with 2 µg of anti-H3K27me3 antibody, 2 µg of anti-H3K27ac antibody or 1.5 µg of anti-flag antibody (Sigma, F1804) combined with 1.5 µg of anti-GRHL2 antibody (antibodies are listed in S11 Table). Protein A/G sepharose beads (GE Healthcare) were added and samples were incubated for 4 h at 4°C under agitation in the presence of 10 µg/mL yeast tRNA (Sigma-Aldrich, R5636). Beads were washed three times with RIPA buffer (50 mM HEPES, 1 mM EDTA, 0.7% Na Deoxycholate, 1% NP40 and 500 mM LiCl) containing 10 µg/mL yeast tRNA and once with TE buffer (10 mM Tris-HCl pH 8.0, 1 mM EDTA). DNA was then eluted in 100 mM NaHCO3 containing 1% SDS and incubated overnight at 65°C for reverse cross-linking. DNA purification was performed using Nucleospin PCR and Gel purification kit (Macherey Nagel, 740609). ChIPed DNA and input samples were subjected to high-throughput sequencing and were further analyzed as described hereafter.

**Histology, immunohistochemistry and immunofluorescent assays**

Liver samples were fixed with 4% paraformaldehyde (PFA; mouse samples) or buffered formalin (human samples) and embedded in paraffin. 4 µm-thick tissue sections were stained with hematoxylin and eosin or underwent an antigen retrieval step with sodium citrate solution (pH 6.0) during 21 min at 110°C into TintoRetriever Pressure Cooker (Bio SB) for immunostaining. Sections were then sequentially incubated with 0.1% Triton X-100 in Tris-Buffered Saline (TBS) for 10 min at 4°C, Bloxall Endogenous Blocking solution (Vector Laboratories) for 10 min at RT, 7% goat serum in 0.05% Tween-20 in TBS (TBS-T) for 15 min at RT, 15 min at RT with 5% milk and 3% BSA in TBS-T and primary GRHL2 antibody (S11 Table) diluted 1:300 in 5% milk and 3% BSA in TBS-T overnight at 4°C. After each step, sections were washed in TBS-T. Subsequently, tissue sections were incubated 30 min at RT with ImmPRESS HRP Polymer anti-mouse IgG Reagent (Vector Laboratories), followed by a TBS wash. Signal was revealed with diaminobenzidine [DAB (Cell Signaling)], and sections were counterstained with hematoxylin (solution Gill N°1, Sigma-Aldrich) and mounted in Aquatex (Sigma-Aldrich). Stained slices were imaged with an Axioscan Z1 slide scanner (Zeiss). Images processing was performed using ZEN software (Zeiss).

For GRHL2 and ALB co-staining, the procedure was adapted as follows. For ALB labeling, instead of using 7% goat serum in TBS-T and DAB, sections were incubated for 15 min at RT with Animal-Free Blocking solution 1× (Cell Signaling) in TBS-T, primary anti-ALB antibody diluted 1:50 (S11 Table), ImmPRESS HRP Polymer horse anti-goat IgG reagent (Vector Laboratories) and 10 min at RT with Alexa Fluor 594 Tyramide SuperBoost reagent (Invitrogen) according to the manufacturer's instructions. Following ALB labeling, similar steps were repeated for GRHL2 staining after an antigen retrieval step using primary anti-GRHL2 antibody (1:300), ImmPRESS HRP Polymer horse anti-mouse IgG reagent (Vector Laboratories), and Alexa Fluor 488 Tyramide SuperBoost reagent (Invitrogen). Stained slices were imaged using Axioscan Z1 slide scanner after nuclei counterstaining with DAPI (62248, Thermo Scientific, 1:10,000).

For triple immunofluorescent staining of GRHL2, ALB, and KRT7, the procedure was further adapted. After blocking with Animal-Free Blocking solution 1× in TBS-T, sections were incubated with primary anti-KRT7 antibody diluted 1:50 (S7 Table), ImmPRESS HRP Polymer horse anti-mouse IgG reagent and Alexa Fluor 488 Tyramide SuperBoost reagent. GRHL2 detection was then performed following an antigen retrieval step, using the same HRP-base amplification system (primary anti-GRHL2 antibody 1:300 and Alexa Fluor 594 Tyramide SuperBoost dye). Finally, ALB staining was carried out using the primary anti-ALB antibody (1:50), followed by Alexa Fluor 647-conjugated donkey anti-goat IgG (A21447, Invitrogen, 1:250). Nuclei were counterstained with DAPI and sections were scanned using an Axioscan Z1 slide scanner.

BMEL cells grown in Matrigel for 4 days were fixed using 4% PFA during 20 min at RT, washed with PBS and quenched with NH₄Cl 50 mM 15 min at RT. Cells were then permeabilized with 0.1% Triton X-100 for 5 min and samples were incubated for 1 h at RT with a blocking solution [PBS + BSA 1%] followed by 1 h incubation with phalloidin (Invitrogen AlexaFluor Plus A30107, 2379366) diluted 1:1000 and 5 min with Hoechst diluted 1:5,000 in PBS + BSA 1%. Samples

were washed with PBS between each step. Cell aggregates were imaged using Celldiscoverer 7 (Zeiss) or confocal microscopy (Zeiss, LSM700) and analyzed using the ZEN software.

## Animal studies

Male mice (C57BL/6J) were purchased from Charles River at 6–11 weeks of age and housed in standard cages in a temperature-controlled room (22–24°C) with 12 hours light-dark cycles and provided with water and standard diet ad libitum. Mice were allowed to acclimate for at least 1 week prior to any experiment.

Hydrodynamic plasmid transfection was performed using intravenous tail vein injections (5-s injections) of 30 μg of plasmid diluted in saline solution (0.9% w/v sodium chloride; final volume was adjusted to mouse body weight, i.e., 0.1 mL/g was used). After hydrodynamic injection, mice were closely monitored during 2 hours. Livers were collected 48 hours later.

Intravenous viral injections using a 26G syringe were performed via the retro-orbital sinus with $4 \times 10^{10}$ viral genomes of AAV8 suspended in 100 μL of sterile 0.9% NaCl per animal. Recombinant AAV8 vectors enabling the expression of GRHL2 (NM_024915.4) under the *EF1A* promoter, or GFP-expressing control viruses, were used (VectorBuilder). Tissue samples were collected 1 or 2 weeks after injection for subsequent analyses.

## Human samples

Liver samples with alcohol-related decompensated cirrhosis were obtained from the TargetOH cohort ("Comparison of Inflammatory Profiles and Regenerative Potential in Alcoholic Liver Disease"; ClinGov NCT03773887; and DC-2008-642) [14,106,107]. Controls consisted in liver samples obtained from the non-cancerous part of liver resections performed in patients with hepatic tumors. All human liver samples were obtained from patients who underwent liver transplantation at Huriez Hospital's Liver Unit (Lille, France). The liver samples were immediately fixed for histology or frozen for RNA and protein extraction.

## Human liver organoid culture and AAV transduction

Human liver organoids were derived from histologically normal liver tissue obtained from patients undergoing PHx for metastasis excision. Tissue samples were collected from macroscopically healthy regions distant from metastatic lesions. Organoid isolation, expansion, and differentiation were performed as described in [77,78]. Briefly, liver cells were dissociated through enzymatic digestion and EPCAM-positive cells sorted out using magnetic-activated cell sorting using anti-EpCAM microbeads (Miltenyi Biotec, 130-061-101) according to the manufacturer's instructions. Cells were embedded in 20 μL domes of growth factor–reduced Matrigel (Corning, 356231) and cultured for two weeks in HepatiCult Organoid Growth Medium (STEMCELL Technologies, 100-0385) to enable organoid formation and expansion. Expanded organoids were dissociated using gentle trypsinization and re-seeded at a density of 1,000 cells per Matrigel dome. After reformation, organoids were transduced with recombinant AAV serotype 3 (AAV3) vectors encoding either GRHL2 (NM_024915.4) or GFP under the *EF1A* promoter (VectorBuilder). Transduction was performed in differentiation medium at a dose of $1 \times 10^{5}$ viral genomes (vg) per organoid-forming cell. Medium was change on the third day post-infection. Organoids were maintained in differentiation medium for an additional 12 days before being harvested and processed for total RNA extraction.

## Origin and recovery of public datasets

Public transcriptomic and functional genomics data used in this study were downloaded from Gene Expression Omnibus (GEO, https://www.ncbi.nlm.nih.gov/geo/) [108], ENCODE [109], Roadmap epigenome (http://www.roadmapepigenomics.org) [110], CistromeDB (http://cistrome.org/db/#/) [111], or the Human Protein atlas (https://www.proteinatlas.org) [37]. Details are provided in S1 and S4 Tables.

The list of mouse TF encoding genes was obtained from the AnimalTFDB 3.0 database [112].

Gene transcriptional start sites (TSS) used in this study were obtained from [113]. For each individual gene, the most active TSS in the mouse liver was considered. Mouse TSS coordinates (mm10; coordinates corresponding to windows of ± 500 bp around the TSS) were used in the LiftOver tool of the UCSC genome browser (https://genome.ucsc.edu/cgi-bin/hgLiftOver) [114] to get the corresponding human coordinates (hg38).

The Epith-Signature gene list was obtained from [36] by selecting "Epithelial" as the cell type in their S7 Table (https://genome.cshlp.org/content/30/7/1047/suppl/DC1). Ensembl ID were converted to official gene symbols using the ENSEMBL Gene ID Converted from biotools.fr (https://www.biotools.fr/human/ensembl_symbol_converter).

Hepatocyte- and cholangiocyte-enriched genes were retrieved from the Human Protein Atlas website (https://www.proteinatlas.org) [34].

### RNA-seq data processing and transcriptomic analyses

**Bulk RNA-Seq data analyses.** For public transcriptomic data, SRA files were first downloaded and converted into FastQ files using the SRA toolkit [115]. FastQ files were uploaded and analyzed on a local instance of Galaxy [116]. Reads were mapped either to the human genome (hg38) or to the mouse genome (mm10) using Hisat2 (version 2.0.1) [117]. Read counting was performed using the count function of the HTSeq tool (version 0.9.1) using the following parameters: mode = UNION, Stranded = No, Minimum alignment quality = 10, Feature types = Exon, ID attributes = Gene ID [118]. Normalization of gene expression data and differential expression analyses were performed in R [119] using the DESeq2 (version 1.42.1) package [120]. Pre-filtering was applied to remove genes with low counts, i.e., genes with cumulated reads in all samples below 10. Significant changes in gene expression were defined considering corrected $p$-values using cut-offs indicated in figure legends. Fold Changes used in our analyses were those provided by DeSeq2. Normalized gene expression data were obtained by dividing Rlog normalized gene expression data obtained from DESeq2 by transcript lengths. Transcript lengths had previously been calculated by averaging the lengths of all transcripts associated to a given gene symbol in Ensembl (GRCm39 or GRCh.38.p13) [121].

Results of bulk RNA-seq transcriptomic data analyses were visualized using dot plots, box plots or pie charts generated using the ggplot2 R package (version 3.5.1) [122] or using heatmaps generated using the ComplexHeatmap R package (version 2.15.4) [123].

S12 Table provides the list and version of all bioinformatical tools used in this study.

**Comparison of transcriptomes using principal component analysis (PCA).** The transcriptome of hepatocytes undergoing HCT (denoted as "reprogrammed hepatocytes" in the initial study) in livers of mice treated with DDC was compared to that of hepatocytes and cholangiocytes from control mice [25]. All genes, pending their expression was detected in all datasets, were considered in these analyses. In order to strengthen the robustness of our analyses, transcriptomes of hepatocytes and cholangiocytes sorted from healthy mouse livers obtained in our laboratory were also considered [33]. Batch correction was performed using the Combat function of the sva R package (version 3.50.0) [124] using the transcriptome of hepatocytes from [33] as the reference dataset and the following parameters: mean. only = T, par.prior = T, prior.plots = F. Next, a PCA was computed using the PCA function of the FactomineR package [125] on the transcriptome of hepatocytes and cholangiocytes from [33], the other datasets being considered as supplemental individuals. The 2 first principal components were used to project individual datasets showing that healthy hepatocytes from the two initial different studies were overlapping. This entire process was similarly performed using only TF encoding genes.

**Gene ontology and pathway enrichment analyses.** Enrichment analysis for biological or molecular pathways were performed using Metascape (https://metascape.org) [126] or Database for Annotation, Visualization, and Integrate Discovery (DAVID 2021, DAVID Knowledgebase v2023q4) [35]. For analyses using DAVID, the top 3,000 deregulated genes were selected based on corrected $p$-values when required. Functional annotation clustering was performed using

default parameters except for similarity term overlap and similarity threshold which were set to 5 and 0.5, respectively. In order to name the clusters, enriched GO terms and pathways from each cluster were used for word counting using WordArt (https://wordart.com).

**Gene set enrichment analyses.** Gene set enrichment analyses (GSEA) were performed using the GSEA software (version 4.3.2) from the Broad Institute [127] and gene sets from S2 Table. We used 1,000 gene-set permutations, no collapse, and the following additional parameters: enrichment statistic = weighted, gene ranking metric = difference of classes. Ranking was performed by the GSEA software using Rlog normalized gene expression data obtained from DeSeq2. In addition to enrichment plots, figures also provide NES and *q*-value, which are the normalized enrichment score and the false discovery rate provided by the GSEA software, respectively.

### Mouse liver single-cell RNA-seq analyses

**Pre-processing of individual datasets.** Data were analyzed with R (version 4.3.1) using raw count matrices obtained as.rds files. Details about all datasets re-analyzed are provided in S1 Table. First, each dataset was filtered using Seurat (version 4.0.3) [128] to keep only cells with more than 1,000 and less than 5,000 expressed genes (excepted for GSE186554 for which the authors had already selected cells comprising 1,500–4,500 expressed genes) and with less than 20% of mitochondrial counts. Details of the Seurat functions used together with all parameters can be retrieved in the supplemental Rscript "1_PreprocessingExample.R" (available at https://zenodo.org/records/13897656), which provides the code used for one of the re-analyzed study. Then data were normalized with sctransform [129] followed by a PCA used to compute UMAP. When required, Louvain clustering was performed in order to remove any remaining cell cluster not within the hepatocyte/cholangiocyte lineages. This involved monitoring expression of the following marker genes: *Alb* and *Apoc3* for hepatocytes, *Sox9* and *Epcam*, for cholangiocytes, *Hnf4a*, *Afp*, *Dlk1*, *Cd13*, *Fgb*, *Fxyd1* and *Gjb1, Id3* for hepatoblast, *Gypa*, *Vim*, *Flt1*, *Pecam1*, *Itgam*, *Clec4f*, *Eln*, *Cd248*, *Upk1b*, *Dcn*, *Lrat*, *Des*, *Pdgfrb*, *Lox*, *Acta2*, *Col1a1* for non-parenchymal cells (NPC). Clusters characterized by expression of NPC marker genes were discarded (see S1 Table for details regarding clustering resolution and numbers of selected cells for each dataset). Finally, to prevent potential bias arising from an unbalanced cell number across datasets and experimental conditions, two datasets were subsampled so that cells from injured livers did not exceed the maximal number of healthy hepatocytes provided by a single study (*n* = 8,964) (S1 Table).

**Integration of the different datasets.** Data were log-normalized and the 2000 most variable genes in each dataset were defined with the vst method using Seurat. Next, STACAS (version 2.0) [130] was used on the 500 common most variable genes selected by SelectIntegrationFeatures function with nfeatures = 500. The FindAnchors and IntegrateData functions were used with the following parameters: dims = 1:30, alpha = 0.8, coverage = 0.5, semisupervised = false. Efficient batch correction was validated on basis of the LISI and ASW scores following guidelines from STACAS. To evaluate integration with the ASW score, a unifying nomenclature where cells were labeled using cell type identity and experimental condition (e.g., healthy control, DDC etc) was used. All codes and parameters are provided in "Rscript 2_Integration_and_Focus.R" available at https://zenodo.org/records/13897656.

**Selection of cells to specifically investigate HCT and trajectory analyses.** Finally, PCA, UMAP, and Louvain clustering were computed on the integrated data provided by STACAS. Data were visualized using ISCEBERG [Interactive Single Cell Expression Browser for Exploration of RNAseq data using Graphics; [131]]. Cells from clusters 2, 8, and 15 (resolution = 0.8; S1B Fig) were further considered after removal of studies contributing less than 5% of all cells from these 3 clusters. In order to compute RNA velocity, healthy hepatocytes from [26], condition: « control »] were added to the selected cells. Reads from all selected cells were mapped to Ensembl (GRCm39 v108) using salmon-alevin (salmon version 1.10.1) [132] to generate spliced and unspliced transcript count matrices. These matrices were batch corrected to match the aforementioned STACAS computed integration. RNA velocity was next computed using ScVelo (v0.2.5 under Python 3.9.5) [133] and data uploaded into CellRank2 (version 2.0.0) [32] to compute cell fate probabilities.

Pseudotime analyses were performed using slingshot (version 2.10.0) [134] and next also uploaded to CellRank2 in order to specifically select cells defining a trajectory of HCT. The batch corrected gene expression data from cells belonging to this HCT trajectory from DDC-treated mice [25] were used to plot gene expression along pseudotime. Normalization was performed with the "LogNormalize" function of Seurat (i.e., for each cell, gene counts were divided by the total number of counts, multiplied by a scale factor set to 10000 and then log-transformed). All codes and parameters are provided in "Rscript 3_Velocity_CellRank.R" available at https://zenodo.org/records/13897656. Gene set expression analyses were performed using the AddModuleScore function of Seurat in ISCEBERG.

## Human single-nuclei RNA-seq data analyses

Data from [11] were mined using the R Shiny application provided by the authors at https://www.mohorianulab.org/shiny/vallier/LiverPlasticity_GribbenGalanakis2024/ or using Seurat and data downloaded from GEO (GSE202379). In all instances, only cells labeled as hepatocytes were considered. Odds ratios and gene expression were computed using hepatocytes originating only from the right lobe so that each patient contributed only one sample to the dataset. Odds ratio were computed using median-unbiased estimation thanks to the oddsratio function of the epitools (version 0.5-10.1) [135].

## Mining transcriptomic data of human primary cells from the Human Protein Atlas

Data from scRNA-seq were downloaded from the Human Protein Atlas website (https://www.proteinatlas.org/about/download) and corresponded to normalized gene expression in transcripts per million (nTPM) issued from pseudo-bulk analyses [37]. Gene expression in different cell lineages (epithelial, mesenchymal, endothelial, etc.) was computed as the mean of nTPM from the individual cell types associated to the different cell lineages according to the nomenclature provided by the Human Protein Atlas.

The Tau index was computed to assess gene expression lineage-specificity using tspex (https://tspex.lge.ibi.unicamp.br) [136] after expression data had been $log_2$-transformed. Expression data for cell lineages were here defined as the median expression from all individual cell types linked to a given lineage.

Hierarchical clustering based on expression of the Epith-Signature was performed using the hclsust (version) and dist functions of the R package stats version 4.3.1 [119] with the following method parameters: Ward.D2 and Euclidean.

## ChIP-Seq data analysis

After initial quality controls of FastQ files, reads were mapped to hg38 using Chromap version 0.1.3-R256 [137] with preset parameters for "chip" including: remove PCR duplicates; error threshold = 8; min-num-seeds = 2; max-seed-frequency = 500−1,000; max-num-best-mappings = 1; max-insert-size = 1,000; MAPQ-threshold = 10; min-read-length = 30; bc-error-threshold = 1; bc-probability-threshold = 0.90.

For H3K27ac and H3K27me3 ChIP-seq, signal tracks were obtained using the bamCoverage function of the DeepTools version 3.5.1 [138] with the following parameters: bin size = 25; normalize = CPM; read extension = 200.

For GRHL2 ChIP-seq, Bam files (n = 4 biological replicates) were analyzed using the Irreproducible Discovery Rate (IDR), which allows to identify TF binding peaks that are reproducible and rank-concordant across replicates (https://github.com/nboley/idr). This was performed using the IDR pipeline from ENCODE (version 2.2.1) installed from https://github.com/ENCODE-DCC/chip-seq-pipeline2/tree/master [139]. ChIP-seq data from mocked transfected cells were used as controls. Since all replicates passed the reproducibility tests of IDR, the top 10,000 peaks based on the IDR score and the average signal from the 4 replicates were used in subsequent analyses. GRHL2 target genes and their enrichment for specific biological terms or pathways were predicted using the CistromeGO tool [69]. The solo mode or the ensemble mode, i.e., combined prediction of target genes using both the GRHL2 binding peaks and genes modulated by GRHL2 as defined using RNA-seq were used with a half-decay distance of 1 kb automatically defined by the CistromeGO algorithm.

All ChIP-seq and chromatin accessibility signals analyses were made looking at gene TSS and monitored using the DeepTools version 3.5.1 [138]. First, the compute matrix tool was used with the following parameters: TSS set as the reference point, binSize of 500 bp, ± 2,500 bp window around the TSS. Then, issued matrices were used to generate average signal plots using the PlotProfile tool. ChIP-seq signals at individual genes were visualized using the Integrated Genome Browser (IGB) (version 10.0.1) [140] or the Integrative Genomics Viewer (IGV) (version 2.17.4) [141].

In order to identify groups of genes with distinct epigenetic patterns around their TSS, the SPARK software (version 1.3.0) was used [42]. SPARK clustering was performed using a window of 2.5 kb centered around gene TSS split into 10 bins and signal normalization set as global, i.e., signal across the whole genome is considered for normalization.

## Statistical analyses

Statistical analyses were performed using R 4.3.1 [119] or GraphPad Prism. The number of samples and statistical tests used are indicated in the figure legends. Corrections for multiple testing was performed by computing false discovery rates reported as $q$-values. Exact $p$- or $q$-values are indicated unless they exceed 0.15. All bar graphs show means ± SD (standard deviations). Box plots are composed of a box from the 25th to the 75th percentile with the median as a line and min to max as whiskers.

## Supporting information

**S1 Fig. Additional characterization of the mouse parenchymal liver cells scRNA-seq atlas and of the cellular trajectory involved in HCT.**
(PDF)

**S2 Fig. Additional characterization of the gene transcriptional changes occurring in HCT.**
(PDF)

**S3 Fig. Additional characterization of deregulated TF-encoding genes during HCT.**
(PDF)

**S4 Fig. Average DHS-seq, H3K36me3 and RNAPolII ChIP-seq signals at the promoters of genes from clusters C1–3 in the mouse liver.**
(PDF)

**S5 Fig. Role of H3K9me3 in the repression of HCT genes in normal hepatocytes.**
(PDF)

**S6 Fig. Basal expression of TF-encoding genes from the C1–3 clusters.**
(PDF)

**S7 Fig. Additional validation of GRHL2 ectopic expression and characterization of induced transcriptional changes in BMEL and HepG2 cells.**
(PDF)

**S8 Fig. Additional representative images of BMEL cell aggregates.**
(PDF)

**S9 Fig. Dose-dependent induction of GRHL2 target genes in HepG2 cells.**
(PDF)

**S10 Fig. Additional validation of GRHL2 ectopic expression in the mouse liver of mice subjected to hydrodynamic tail vain injection.**
(PDF)

**S11 Fig. Additional immunostainings of GRHL2 in human end-stage chronic liver diseases.**
(PDF)

**S12 Fig. Characterization of GRHL2-expressing hepatocytes in human MASLD using snRNA-seq data.**
(PDF)

**S13 Fig. Additional immunostainings of GRHL2 in human PBC and PSC samples.**
(PDF)

**S1 Table. Details regarding single-cell RNA-seq datasets used in Fig 1.**
(XLSX)

**S2 Table. Gene sets used in this study.**
(XLS)

**S3 Table. List of genes and TF-encoding genes differentially expressed between HCT and Hepatocytes.**
(XLSX)

**S4 Table. Transcriptomic and functional genomic data used in this study.**
(XLS)

**S5 Table. List of genes down-regulated upon ectopic expression of GRHL2 in BMEL cells ($q \leq 0.05$).**
(TXT)

**S6 Table. List of genes up-regulated upon ectopic expression of GRHL2 in BMEL cells ($q \leq 0.05$).**
(TXT)

**S7 Table. List of genes down-regulated upon ectopic expression of GRHL2 in HepG2 cells ($q \leq 0.05$).**
(TXT)

**S8 Table. List of genes up-regulated upon ectopic expression of GRHL2 in HepG2 cells ($q \leq 0.05$).**
(TXT)

**S9 Table. Features of the selected GRHL2 target genes.**
(XLSX)

**S10 Table. Primer sequences used in this study.**
(XLS)

**S11 Table. Antibodies used in this study.**
(XLSX)

**S12 Table. Bioinformatical tools used in this study.**
(XLS)

**S1 Data. All individual numerical values that underlie the experimental data generated specifically for this study.**
(XLSX)

**S1 Raw Images. Uncropped images issued from western blotting or simple western immunoassays.**
(PDF)

## Acknowledgments

The authors thank Gilles Salbert (Univ Rennes) and Réjane Paumelle (Univ Lille) for discussions, Jean-Sébastien Annicotte and Joel Haas for help with animal experimentation and Solène Audry for technical assistance. The authors are indebted to Yassine Jarmouni and the BioImaging Center Lille (BICeL; Plateformes Lilloises en Biologie et Santé (PLBS) – UMS 2014 – US 41, Univ. Lille). We acknowledge Bilille platform – US 41 – UAR 2014 – PLBS and the i–tensive computing center of the University of Lille for the computing resources (Cloud Bilille).

## Author contributions

**Conceptualization:** Ludivine Vasseur, Céline Gheeraert, Julie Dubois-Chevalier, Ninon Very, Laurent Dubuquoy, Jérôme Eeckhoute.

**Formal analysis:** Ludivine Vasseur, Céline Gheeraert, Julie Dubois-Chevalier, Ninon Very, Mohamed Bou Saleh, Jérôme Eeckhoute.

**Funding acquisition:** Bart Staels, Philippe Lefebvre, Laurent Dubuquoy, Jérôme Eeckhoute.

**Investigation:** Ludivine Vasseur, Céline Gheeraert, Julie Dubois-Chevalier, Ninon Very, Loïc Guille, Mohamed Bou Saleh, Clémence Boulet, Cyril Sobolewski, Pascal Loyer, Alexandre Berthier, Noémie Legrand, Solenne Taront, Jérôme Eeckhoute.

**Methodology:** Ludivine Vasseur, Céline Gheeraert, Julie Dubois-Chevalier, Ninon Very, Loïc Guille, Mohamed Bou Saleh, Clémence Boulet, Cyril Sobolewski, Pascal Loyer, Alexandre Berthier, Noémie Legrand, Anne Corlu, Viviane Gnemmi, Antonino Bongiovanni, Nicolaj I Toft, Lars Grøntved.

**Project administration:** Bart Staels, Philippe Lefebvre, Laurent Dubuquoy, Jérôme Eeckhoute.

**Resources:** Viviane Gnemmi, Guillaume Lasailly, Emmanuelle Leteurtre, David Tulasne, Alessandro Furlan, Line Carolle Ntandja-Wandji, Laurent Dubuquoy.

**Software:** Ludivine Vasseur, Julie Dubois-Chevalier, Dmitry Galinousky, Antonino Bongiovanni.

**Supervision:** Bart Staels, Laurent Dubuquoy, Jérôme Eeckhoute.

**Validation:** Ludivine Vasseur, Céline Gheeraert.

**Visualization:** Ludivine Vasseur, Céline Gheeraert, Julie Dubois-Chevalier, Ninon Very, Loïc Guille, Mohamed Bou Saleh, Viviane Gnemmi, Dmitry Galinousky, Nicolaj I Toft, Lars Grøntved, Jérôme Eeckhoute.

**Writing – original draft:** Ludivine Vasseur, Jérôme Eeckhoute.

**Writing – review & editing:** Ludivine Vasseur, Céline Gheeraert, Julie Dubois-Chevalier, Ninon Very, Mohamed Bou Saleh, Pascal Loyer, Alexandre Berthier, Anne Corlu, Viviane Gnemmi, Emmanuelle Leteurtre, Bart Staels, Philippe Lefebvre, Laurent Dubuquoy, Jérôme Eeckhoute.

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
