## [Editor Report · Decision Letter 0]

5 Feb 2025

Dear Dr Eeckhoute,

Thank you for submitting your manuscript entitled "Direct hepatocyte-to-cholangiocyte transdifferentiation driven by derepression of the epithelial transcription factor GRHL2" for consideration as a Research Article by PLOS Biology.

Your manuscript has now been evaluated by the PLOS Biology editorial staff as well as by an academic editor with relevant expertise and I am writing to let you know that we would like to send your submission out for external peer review.

Once your full submission is complete, your paper will undergo a series of checks in preparation for peer review. After your manuscript has passed the checks it will be sent out for review. To provide the metadata for your submission, please Login to Editorial Manager (https://www.editorialmanager.com/pbiology) within two working days, i.e. by Feb 07 2025 11:59PM.

Kind regards,

Luke

Lucas Smith, Ph.D.

Senior Editor

PLOS Biology

lsmith@plos.org

---

## [Decision Letter · Decision Letter 1]

3 Apr 2025

Dear Jerome,

Thank you for your patience while your manuscript "Direct hepatocyte-to-cholangiocyte transdifferentiation driven by derepression of the epithelial transcription factor GRHL2" was peer-reviewed at PLOS Biology. It has now been evaluated by the PLOS Biology editors, an Academic Editor with relevant expertise, and by several independent reviewers.

In light of the reviews, which you will find at the end of this email, we would like to invite you to revise the work to thoroughly address the reviewers' reports.

As you will see below, while the reviewers appreciate that your study offers interesting insights, they have also raised a number of important points which we think will need to be addressed before we can consider your study for publication. Reviewers 2 and 3, in particular, identify important limitations with this study, including the use of HepG2 cancer cell lines and limited data demonstrating GRHL2's direct role in the transition process - and we feel these points will need to be experimentally addressed.

We think the revision will need to more clearly demonstrate the mechanistic role of GRHL2 in inducing biliary cell fate while suppressing hepatocyte fate, using a primary organoid model system. Additionally, we think it would strengthen the study to revisit the in vivo mouse model to assess biliary cell fate in GRHL2-overexpressing cells and to test the direct link between EZH2, PRC2, and the modulation of GRHL2 expression. We also agree with reviewer 3 that it would be valuable to discuss your findings in the context of previous studies on hepatocyte-to-cholangiocyte transition (HCT). As a last editorial note, we would not require that you provide additional loss-of-function data as suggested by reviewer 2. We appreciate that that line of research would be interesting, but we do no think it is essential to support the conclusions, if you develop the study as outlined above, and thoroughly address the other reviewer points.

Given the extent of revision needed, we cannot make a decision about publication until we have seen the revised manuscript and your response to the reviewers' comments. Your revised manuscript is likely to be sent for further evaluation by all or a subset of the reviewers.

We expect to receive your revised manuscript within 3 months. However, we understand that it may require a lot of work to expand the mechanistic insights provided by the study and so we would be happy to extend the deadline for the revision by a few months, if needed. Please email us (plosbiology@plos.org) if you have any questions or concerns, or would like to request an extension.

**IMPORTANT - SUBMITTING YOUR REVISION**

*Re-submission Checklist*

*Published Peer Review*

*PLOS Data Policy*

*Blot and Gel Data Policy*

Sincerely,

Luke

Lucas Smith, Ph.D.

Senior Editor

PLOS Biology

lsmith@plos.org

REVIEWS:

Reviewer #1, Holger Willenbring (note, reviewer 1 has signed this review): The manuscript by Vasseur et al investigates epigenetic mechanism of hepatocyte-to-cholangiocyte transdifferentiation, which has been identified as a response to liver injury in both mice and humans. The authors make good use of published datasets to arrive at the compelling conclusion that hepatocyte-to-cholangiocyte transdifferentiation is, at its core, the introduction of a generic epithelial transcriptional program at the expense of the unique transcriptome and specification characteristic for hepatocytes. The authors show that this process does not involve a progenitor-like intermediate as suggested by others. They identify an epigenetic mechanism regulating this process, that is, loss of EZH1/2 expression freeing epithelial transcription factor genes such as GRHL2 from repression by H3K27me3. Their conclusions rest on multiple lines of evidence, including functional testing in several liver cell lines. The data are clearly presented and described, and rounded out by several tables. There are only a couple of places where the descriptions seem incomplete or off.

1. In the results on page 13 it is not totally clear what genes constitute the "cholangiocyte-like transcriptional program", that is, what genes are induced by GRHL2 beyond established downstream targets like RAB25 and CLDN4 and the four others genes that are mentioned. This deficiency could be addressed by highlighting genes in Fig. S7B, D and expanding Supplementary Table 5. Along these lines, the authors mention SOX4 on page 18 but it does not seem to feature in any of the figures.

2. In the discussion on page 18 the authors refer to a paper by Yang et al from 2023 as being "at apparent odds with our findings". However, that paper shows that EZH2 inhibition or knockout results in more EPCAM-positive cells (that is, cholangiocytes) in E12.5 liver slices (Fig. 5B-D) or in mice at E17.5 and P0 (Fig. 6A-D), which is associated with reduced H3K27me3 levels and therefore seems consistent with the results by Vasseur et al.

Reviewer #2, Pinglong Xu (note, reviewer 2 has signed this review): The manuscript titled "Direct hepatocyte-to-cholangiocyte transdifferentiation driven by derepression of the epithelial transcription factor GRHL2" by Eeckhoute et al. presents interesting findings that adult hepatocytes can convert directly into cholangiocytes (HCT) by relieving polycomb-mediated epigenetic repression of epithelial transcriptional regulator GRHL2. The authors propose that HCT bypasses a hepatoblast-like state, instead triggering a monopolarized epithelial gene program upon GRHL2 induction, thereby suggesting that derepressing an epithelial transcription factor is sufficient for lineage reprogramming and providing implications in human liver diseases.

Understanding HCT is crucial for liver regeneration and for understanding chronic liver diseases. These findings highlight the importance of the repression of the epithelial transcription program during the HCT transition. The manuscript presents a convincing data analysis supporting the conclusions drawn. However, a few key issues remain unaddressed:

1. The manuscript provides strong gain-of-function evidence that overexpression of GRHL2 in hepatocytes can induce epithelial gene expression and promote HCT both in vitro and in vivo. However, the proposed mechanism remains incomplete due to the lack of rigorous loss-of-function data. Will depleting or inhibiting GRHL2 interrupt the HCT process, such as during the in vitro HCT system and with a hepatocyte-specific knockout of GRHL2 by AAV or other systems?

2. How precisely PRC2 and H3K27me3 target GRHL2 and other "epithelial" genes in hepatocytes is unclear. Is there direct evidence that modulating EZH2 or PRC2 activity can promptly derepress GRHL2?

3. Beyond changes in transcript profiles, do these cells acquire bile duct-like structures or secretory functions to confirm full cholangiocyte functionality?

4. Fig. 6B, the correlation between GRHL2 protein levels and H3K27me3/EZH2 is weak in individual samples.

Reviewer #3: This study aims to further understand the mechanisms controlling the transdifferentiation of hepatocytes into cholangiocytes (or so called HCT by the authors) which has been well studied in the DDC mouse model and more recently described in human liver affected by progressive MASLD. Analysing published data set, the authors confirm that HCT does not involve a intermediate progenitors state and they also uncover GRHL2 as a potential transcription factor regulating this process. These data are interesting and are in line with previous studies. The main interesting part is the role of GRHL2 and additional validations are necessary to validate the conclusions of this study.

HCT has been widely studied in the DDC mouse model. These previous studies have already uncovered key epigenetic regulations and regulators such as Sox4 [ doi: 10.1038/s41467-024-45939-z.]. Similarly, a recent publication in human have demonstrated the existence of HCT in chronically injured liver and already established the absence of progenitor state [DOI: 10.1038/s41586-024-07465-2]. It would be very useful that the authors discussed their results in more details in the context of these previous publications.

The epigenetic analyses (ChIP-Seq) are performed on very heterogeneous population. The hepatocytes represent 80% of the liver mass. However, they are extremely heterogenous due to zonation. This heterogeneity could play a key role in regenerative capacity and thus in the interpretation of their analyses.

The characterization of GRHL2 function is too limited and not convincing. Indeed, the data presented show that GRHL2 manipulation induces an epithelial signature but not a cholangiocytes signature. Indeed, they don't show the expression of cholangiocytes markers especially KRT19, KRT7, BICC1 etc. They need to provide data (QPCR and immunostaining with primary cells control cholangiocytes and hepatocytes) demonstrating that GRHL2 truly induces a change in cellular identity. HEPG2 or other transformed cells are a very poor model for these experiments. Tumor cells are likely impossible to transdifferentiate. They should use cholangiocytes organoids. Finally, the human data need to be reinforced. They need to show that GRHL2 is expressed in biphenotypic cells co expressing KRT19/ALB for example.

Minor comments:

Figure 1F:

* ALB expression should be investigated and shown to support the findings.

* Clarification is needed regarding the type of normalization used. Was normalization performed relative to a control group or endogenous gene expression?

* It would be helpful to indicate the drawn trajectory on the UMAP visualization. Specifically, does the trajectory begin from cluster 4 and cross cluster 1? Is the trajectory determined only by the cells that overlap with cluster 1?

Figure 2

* Have cholangiocytes been traced to confirm they do not transdifferentiate into hepatocytes in this model?

Figure 4

* GRHL2 Overexpression (OE): It should be demonstrated whether GRHL2 OE has any negative effects on hepatocyte differentiation. Additional data on the impact of GRHL2 ectopic expression on hepatocyte signature should be included.

* BMEL Cells: Are genes involved in hepatocyte differentiation downregulated in BMEL cells in concert with GRHL2 OE? This aspect should be investigated and presented.

Figure 5

* Similar to Figure 4, it is necessary to determine and present whether GRHL2 OE affects hepatocyte differentiation and signature-related genes.

---

## [Decision Letter · Decision Letter 2]

11 Nov 2025

Dear Jerome,

Thank you again for your patience while we considered your revised manuscript "Direct hepatocyte-to-cholangiocyte transdifferentiation driven by derepression of the epithelial transcription factor GRHL2" for publication as a Research Article at PLOS Biology. This revised version of your manuscript has been evaluated by the PLOS Biology editors, the Academic Editor and the original reviewers.

You will see that two of the reviewers are fully satisfied by the revision. Reviewer 3 also appreciates that the study has been strengthened after your revision, but s/he has some lingering concerns and suggests that more markers are needed to support claims about cholangiocyte transdifferentiation. Reviewer 3 suggests that you either characterize additional markers, or tone down some of the conclusions (including your title). After discussion with the Academic Editor, we agree with Reviewer 3's lingering concern. We would welcome new data to address his/her comment, but we are also OK with you toning down the conclusions and title of your study to more clearly reflect the limitations of the study.

Based on the reviews, we are likely to accept this manuscript for publication, provided you satisfactorily address the remaining points raised by reviewer 3, with new data or textual changes. Please also make sure to address the following data and other policy-related requests.

**IMPORTANT - Please address the following editorial points:

1) ETHICS STATEMENT: Please update your ethics statement to include the approval number for the protocol approved by the Lille Ethical Committee.

2) DATA: Thank you for providing the CHIP-Seq and RNA-seq data on GEO. Please add a note to every relevant figure legend, pointing readers to these datasets. (ex you can say "the data underlying this figure can be found at ___")

3) DATA: In addition to the data you have already provided, we also need you to provide the underlying data for all other figures in your paper. While the RNA-seq and CHIP-seq data are fine, we also need the underlying data for your qPCR and Western blot analysis, etc (ex Fig 8A-B)

a. Supplementary files (e.g., excel). Please ensure that all data files are uploaded as 'Supporting Information' and are invariably referred to (in the manuscript, figure legends, and the Description field when uploading your files) using the following format verbatim: S1 Data, S2 Data, etc. Multiple panels of a single or even several figures can be included as multiple sheets in one excel file that is saved using exactly the following convention: S1_Data.xlsx (using an underscore).

b. Deposition in a publicly available repository. Please also provide the accession code or a reviewer link so that we may view your data before publication.

4) CODE: Thank you for depositing the code for your study on Zenodo. I see that those files are currently set to private, and I could not access the code (even when I used the reviewer links that you provided). Can you provide me with reviewer access so that I can verify that the code deposition meets our requirements? Note that you will need to make all code and data public before publication.

5) BLOT AND GEL IMAGES: Thank you for providing a file with the raw western blot images for your study. The images that you provided look a bit cropped, and so I ask that you edit this file to include the fully uncropped images taken from the scanner.

We expect to receive your revised manuscript within two weeks.

*Published Peer Review History*

*Press*

Sincerely,

Lucas

Lucas Smith, Ph.D.

Senior Editor

lsmith@plos.org

PLOS Biology

Reviewer remarks:

Reviewer #1: The revised manuscript by Vasseur et al. addresses the points raised in the initial review. The revised manuscript convincingly shows that hepatocyte-to-cholangiocyte transdifferentiation can be driven by derepression of GRHL2 and activation of a canonical epithelial transcriptional program, rather than by reactivating a progenitor-like state. The revised manuscript includes substantial new data that strengthen the mechanistic and functional analyses. The revised manuscript is clearly written, well contextualized, and ready for publication.

Response to previous concerns

1. Definition of the cholangiocyte-like transcriptional program: The authors now define this program as induction of an "Epith-Signature" capturing over 60% of cholangiocyte-enriched genes. They included complete gene lists (Suppl. Table 2), full sets of GRHL2-induced genes (Suppl. Tables 6 and 8), and highlighted direct targets such as ESRP1 and KRT7 (Figs. 6, 7). The role of SOX4 is clarified (p. 18).

2. Clarification of EZH2 effects in the discussion: The confusing statement regarding Yang et al. (2023) has been corrected to state that EZH2 inhibition promotes cholangiocyte differentiation, which is consistent with the authors' findings.

3. Addition of mechanistic and functional evidence: New data show direct binding of EZH2 and H3K27me3 at the Grhl2 promoter (Fig. 5), derepression of Grhl2 after Ezh1/2 loss, and activation of Epith-Signature genes. GRHL2 overexpression promotes 3D cyst formation in BMEL cells and reprogramming of hepatocytes in vivo (Figs. 5-7).

4. Integration with recent literature and human data: The revised discussion references recent studies (SOX4/H3K36me axis, MASLD, spatial transcriptomics) and emphasizes concordance of findings in mice with human alcoholic hepatitis, PSC, and PBC samples showing GRHL2+/KRT7+ hepatocytes at scar regions (Fig. 8).

5. Additional experimental validation: The authors performed new GRHL2 expression experiments in human liver organoids, confirming the induction of epithelial/cholangiocyte genes and repression of hepatocyte markers (Fig. 6J). Zonation and cell-type heterogeneity were examined using snATAC-seq and hepatocyte-specific epigenomic data.

Reviewer #2, Pinglong Xu (note, Reviewer 2 has signed this review): The questions raised during my initial review have largely been addressed and the manuscript is significantly improved.

Reviewer #3: The authors have done a lot of efforts to answer the reviewers' comments. The data showing that GRHL2 plays a role in the transdifferentiationof hepatocytes into cholangiocytes are indeed convincing. However, the problem remains the capacity of GRHL2 to directly drive the transdifferentiation of hepatocyte into cholangiocytes. The functional data either in vitro or in vivo show that GRHL2 overexpression results in the up regulation of epithelial markers which are expressed in a number of cells (EHF, CLDN4). KRT7 and ESPR1 are indeed specific for cholangiocytes. However, KRT7 is associated with functional maturation and KRT19 is more broadly used. What about other specific/functional markers markers which are universally used such as KRT19, HNF1B, CFTR, BICC1, EPCAM, DCD2 etc These markers are commonly used to characterise cholangiocytes. Why are they not using these markers in Figure 5, 6 and 7. It seems that the authors have handpicked few markers which can support their conclusions. Of note the data in human mainly show that GRHL2 is expressed in ductular reaction. These cholangiocytes are usually located closed to the regenerative nodule but this does not mean that they are produced by hepatocytes. Furthermore, only few biphenotypic cells truly express GRHL2. So again GRHL2 is likely to participate in the transdifferentiation process but not to drive it. This factor is not sufficient for this process. So, it would be incredibly useful either to include the markers mentioned above in Figure 6 and 7 or to adapt their conclusions/title to reflect their results more accurately.

---

## [Editor Report · Decision Letter 3]

21 Nov 2025

Dear Jerome,

Thank you for the submission of your revised Research Article "Direct hepatocyte-to-cholangiocyte transdifferentiation promoted by derepression of the epithelial transcription factor GRHL2" for publication in PLOS Biology, and thank you for addressing the last reviewer and editorial requests in this revision. On behalf of my colleagues and the Academic Editor, Joo-Hyeon Lee, I am pleased to say that we can in principle accept your manuscript for publication, provided you address any remaining formatting and reporting issues. These will be detailed in an email you should receive within 2-3 business days from our colleagues in the journal operations team; no action is required from you until then. Please note that we will not be able to formally accept your manuscript and schedule it for publication until you have completed any requested changes.

**IMPORTANT - as you address any production requests, to come, we would also like to propose a tweak to the title, to make it a bit more active. Specifically, we suggest you change the title to:

"Derepression of the epithelial transcription factor GRHL2 promotes direct hepatocyte-to-cholangiocyte transdifferentiation"

PRESS

Sincerely, 

Lucas Smith, Ph.D.

Senior Editor

PLOS Biology

lsmith@plos.org